# Human-AI Collaborative Uncertainty Quantification

Sima Noorani [* 1]  Shayan Kiyani [* 1]  George Pappas [1]  Hamed Hassani [1]

## Abstract

AI predictive systems increasingly support high-stakes decision making, yet robust decisions under uncertainty often rely on human capabilities beyond AI alone. This motivates collaborative approaches that combine human judgment with AI predictions. We study this problem through the lens of uncertainty quantification and introduce *Human-AI Collaborative Uncertainty Quantification*, a framework in which an AI system refines a human expert's proposed prediction set subject to two principles: *counterfactual harm*, requiring that the AI not degrade correct human judgments, and *complementarity*, requiring recovery of correct outcomes the human missed. At the population level, we show that the optimal collaborative prediction set has a simple two-threshold structure over a single score function, governing pruning and augmentation relative to the human proposal. Building on this characterization, we develop offline and online calibration algorithms with *distribution-free* finite-sample guarantees. The online algorithm adapts to arbitrary distribution shifts, including settings where human behavior evolves through interaction with the AI. Empirically, we show that collaborative prediction sets outperform human-only and AI-only baselines, achieving improved coverage–efficiency tradeoffs across image classification, regression, and text-based medical decision making.

## 1. Introduction

Artificial intelligence has demonstrated extraordinary predictive power, enabling data-driven decision-making in high-stakes domains such as healthcare, law, and autonomous systems. These systems excel at extracting patterns from vast amounts of data, offering statistical accuracy and consistency at a scale unattainable by human reasoning alone. Yet, robust decision-making in such settings requires more than predictive accuracy. Human experts contribute domain knowledge beyond data (Hansen & Quinon, 2023), persistent memory for long-term planning and context (Bengio et al., 1994), and the ability to reason and act within the physical world in ways still inaccessible to current AI systems (Agrawal, 2010). These complementary strengths point to the importance of human-AI collaboration, where computational precision and human judgment can jointly guide decisions under uncertainty.

A central challenge in realizing this vision lies in uncertainty quantification (UQ). Precise characterization of uncertainty is fundamental to robust decision-making, as it allows decision-makers to weigh risks, assess reliability, and allocate trust between human and machine. While UQ has been extensively studied in the machine learning community, these efforts largely focus on AI systems in isolation. In collaborative settings, however, it is not clear what principles of UQ should be when humans and AI are jointly in the loop. Identifying these principles is essential for designing frameworks that achieve the best of both worlds: combining AI's predictive accuracy with human judgment to enable decisions more robust and effective than either could do alone. To this end, we ask:

> *What should characterize a successful collaboration between a human expert and an AI system?*

Two principles naturally emerge. First, the expert must trust the collaboration to be willing to engage: the AI should not degrade the quality of the human's input. In other words, collaborating with AI should not make the outcome worse in the worst case—a notion we refer to as *counterfactual harm*. Second, collaboration must offer clear benefits beyond what the expert could achieve alone. The AI should *complement* the human by addressing blind spots, identifying correct outcomes that may have been overlooked, and thereby strengthening the overall decision process. Together, these two principles, trust through non-degradation and benefit through complementarity, capture the essential properties of a meaningful human–AI collaborative framework.

In this work, motivated by recent advances in conformal

[1]Department of Electrical and Systems Engineering, University of Pennsylvania, USA. Correspondence to: Sima Noorani <nooranis@seas.upenn.edu>, Shayan Kiyani <shayank@seas.upenn.edu>.

*Proceedings of the 43rd International Conference on Machine Learning*, Seoul, South Korea. PMLR 306, 2026. Copyright 2026 by the author(s).

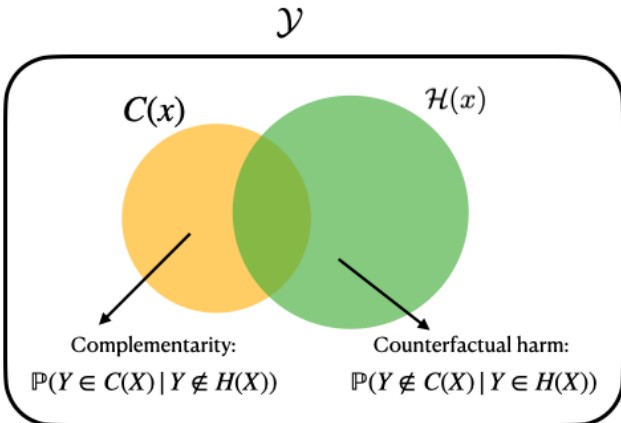

*Figure 1.* Schematic of the two guiding principles

prediction (Vovk et al., 2005; Lei et al., 2017; Romano et al., 2019; 2020; Angelopoulos et al., 2022), we develop a framework that instantiates these two principles in the context of collaborative prediction sets. This allows us to design distribution-free sets that respect both principles without assumptions on the behavior of the AI model or the human, making the approach practical for modern applications. Additionally, recent work shows that conformal prediction sets are essential for risk-sensitive decision making, where decisions must account for predictive uncertainty in a principled way (Kiyani et al., 2025). This makes prediction sets an especially compelling subject of study for human–AI collaboration in high-stakes domains such as healthcare.

**Proposed Framework.** We study *human–AI collaborative uncertainty quantification*, where the two agents jointly construct a prediction set. Formally, let $(X, Y) \sim \mathcal{P}$, where $X \in \mathcal{X}$ denotes the observed features and $Y \in \mathcal{Y}$ the corresponding label. The goal is to construct, for each input $x$, a set $C(x) \subseteq \mathcal{Y}$ that contains the true label $Y$ with high probability while remaining as small as possible.

In our collaborative setting, a human expert first proposes an initial set of plausible outcomes $H(x) \subseteq \mathcal{Y}$. The AI system then refines this proposal by outputting a prediction set $C(x, H(x)) \subseteq \mathcal{Y}$, designed to complement the human input. For notational convenience, we drop the explicit dependence on $H(x)$ in what follows, and have $C(x) := C(x, H(x))$.

This modification is guided by two principles. The first is *low counterfactual harm*: the AI should not degrade the quality of the human proposal. Concretely, whenever the true label lies within the human's proposed set, the AI's refinement must preserve high coverage,

$$\mathbb{P}(Y \notin C(X) \mid Y \in H(X)) < \varepsilon. \quad (1)$$

The second is *complementarity*: the AI should add value precisely when the human misses the correct outcome. That

is, with high probability, the AI's refinement recovers the true label whenever it is excluded from the human proposal,

$$\mathbb{P}(Y \in C(X) \mid Y \notin H(X)) \geq 1 - \delta. \quad (2)$$

These two principles are illustrated schematically in Figure 1. Together, they formalize a collaborative prediction strategy: the AI preserves the human's expertise while compensating for potential blind spots. They come together in the following optimization problem, which serves as the collaboration framework we study in this work:

---

**Human-AI Collaboration Optimization (HACO)**

$$\min_{C:\mathcal{X}\to 2^{\mathcal{Y}}} \quad \mathbb{E}\,|C(X)| \quad \text{(HACO)}$$

$$\text{s.t.} \quad \mathbb{P}(Y \notin C(X) \mid Y \in H(X)) < \varepsilon,$$

$$\mathbb{P}(Y \in C(X) \mid Y \notin H(X)) \geq 1 - \delta,$$

where $\varepsilon$ and $\delta$ are two user-defined thresholds.

---

At a high level, the goal of prediction sets is to include the correct label with high probability while keeping the sets small — set size serving as the measure of efficiency in uncertainty quantification. Within our framework, the AI contributes in two complementary ways: pruning and augmentation. On the one hand, the AI prunes labels from the human proposal whenever possible, since smaller sets are more informative, but does so without violating the counterfactual harm constraint. On the other hand, the AI augments the set by adding likely labels that the human may have overlooked, thereby ensuring complementarity. The human contribution, in turn, is to provide the AI with a stronger starting point. When the initial human-proposed sets are of high quality, the AI's final sets achieve the same coverage level with significantly smaller size than what either could have produced in isolation.

**Preview of Results.**

1) We characterize the optimal solution to HACO in Section 4. As we will show, the optimal solution takes the intuitive form of "two thresholds over one score function", one threshold for pruning labels in the human set, and the other guides the labels that we will add to the human set. We will then build upon this result to design conformity scores that will be used by our finite sample algorithm. In particular, for the case of regression, our score is a novel extension of conformalized quantile regression (Romano et al., 2019).

2) In Section 4, we derive practical finite sample algorithms with provable distribution-free guarantees, in two settings of offline, where the calibration and test data are separated and exchangeable, and online, where the data is streamed one by one. Notably, in the online setting, our algorithm also captures the novel concept of "Human-to-AI Adaptation", which might be of its own interest.

3) In Section 5, we evaluate our finite sample offline and online algorithms on three data modalities: image classification, text based medical diagnosis, and real-valued regression. Across all settings, we show that the parameters $\varepsilon$ and $\delta$ can be tuned such that the collaborative prediction set outperforms both human and AI-only baselines, achieving higher coverage, smaller size, or both. We vary human and AI strength to study each component's role and test robustness under various distribution shifts.

## 1.1. Related Works

We briefly discuss closely related works here and defer a full discussion to Appendix A. In the context of the human–AI collaboration, a growing line of work studies prediction sets as advice to experts (Straitouri & Rodriguez, 2024; Straitouri et al., 2023; Cresswell et al., 2024; Zhang et al., 2024; Paat & Shen, 2025). For instance, Straitouri et al. (2023) propose improving expert predictions with conformal prediction sets, Babbar et al. (2022) show empirically that set-valued advice can boost human accuracy, and Straitouri et al. (2024) analyze such systems through the lens of counterfactual harm. These works differ from ours in that they study how humans use AI-provided sets and evaluate downstream human accuracy, but do not construct a final *collaborative* prediction set that algorithmically integrates human feedback with AI. A complementary literature on *learning to defer* allocates instances between models and experts (Madras et al., 2018; Mozannar & Sontag, 2021; Okati et al., 2021; Verma & Nalisnick, 2022). This also differs from our goal in that we do not optimize who decides on each instance; instead, we collaboratively quantify uncertainty by combining the human's initial set with AI to return a single, joint prediction set with explicit safeguards (e.g., counterfactual harm and complementarity constraints).

## 2. Optimal Prediction Sets over Population

We begin by characterizing the optimal solution to the optimization problem HACO, the problem introduced in Section 1, in the infinite-sample regime, where the data distribution $\mathcal{P}$ is fully known. This characterization uncovers the statistical framework that we will later use to design finite-sample algorithms, enabling us to tune the dynamics of Human-AI collaboration with fine control over counterfactual harm and the complementarity rate of the collaboration procedure.

**Theorem 2.1.** *The optimal solution to HACO is of the form* $C^*(x) = \{ y : 1 - p(y \mid x) \leq a^* \mathbf{1}\{y \notin H(x)\} + b^* \mathbf{1}\{y \in H(x)\}\}$, *a.s. for any* $x \in \mathcal{X}$, *for thresholds* $a^*, b^* \in \mathbb{R}$.

The theorem shows that the optimal collaborative prediction set can be described by two thresholds: One, $b^*$, which is responsible for *pruning*, i.e., for the labels $y \in H(x)$, $b^*$ determines which ones we keep and which ones we exclude;

And the other, $a^*$, which is responsible for *augmenting* new labels, i.e., for the labels $y \notin H(x)$, $a^*$ determines which ones to add to the final set.

This theorem generalizes prior results on minimum set size conformal prediction (Sadinle et al., 2019; Kiyani et al., 2024). When the human set always includes all the labels or is always empty—essentially the two cases in which the human set carries no information about the true label—the optimal set reduces to a one-scalar characterization of the form $\{ y : 1 - p(y \mid x) \leq q^* \}$, which corresponds to minimum set size conformal prediction.

In what follows, we take advantage of the result of this theorem to design an algorithmic framework for Human-AI collaboration. In particular, in the characterization given by Theorem 2.1, there are three components that need to be approximated or estimated with finite samples: $p(y \mid x)$, $a^*$, and $b^*$. The AI's role will be to provide an approximation of $p(y \mid x)$. In the next section, we will discuss this in both classification and regression. We will then discuss debiasing strategies to estimate $a^*$ and $b^*$ from.

## 3. Conformal Scoring Rules

Building on the results of Theorem 2.1, our goal is to construct prediction sets of the form $C^*(x) = \{ y : s(x,y) \leq a^* \mathbf{1}\{y \notin H(x)\} + b^* \mathbf{1}\{y \in H(x)\} \}$, where $s(x,y)$ is a **non-conformity score** that measures how unusual a label $y$ is for a given input $x$. In the infinite-sample regime, Theorem 2.1 shows that the optimal non-conformity score is $s(x,y) = 1 - p(y \mid x)$ where $p(y \mid x)$ is the true conditional distribution. However, since $p(y \mid x)$ is unknown in practice, we design a score to approximate the behavior of the optimal score. Below, we describe how such scores can be constructed for both classification and regression settings.

**Classification** In classification tasks, predictive models typically output a probability distribution over labels, often obtained via a softmax layer. Formally, let $f : \mathcal{X} \to \Delta_{\mathcal{Y}}$ map each input $x \in \mathcal{X}$ to a $|\mathcal{Y}|$-dimensional vector of probabilities $\hat{p}(y \mid x)$, which approximates the true conditional probabilities $p(y \mid x)$. A widely used non-conformity score in classification (Sadinle et al., 2019) that we adopt in our framework is defined as $\hat{s}(x,y) = 1 - \hat{p}(y \mid x)$,

**Regression** In regression, the continuous label space makes it difficult to estimate the full conditional distribution $p(y \mid x)$, so directly approximating the optimal score $1 - p(y \mid x)$ is not straightforward. To circumvent this, we build upon *Conformalized Quantile Regression* (CQR) (Romano et al., 2019). The idea of CQR is to estimate lower and upper conditional quantiles of $Y$ given $X = x$ and then use them to construct a conformal score. Suppose we obtain an estimate $\hat{q}_{\alpha/2}$ of the $\alpha/2$ quantile of

the distribution of $Y \mid X = x$, and an estimate $\hat{q}_{1-\alpha/2}$ for the $1 - \alpha/2$ quantile. We can then define the score $\hat{s}(x, y) = \max\left\{\hat{q}_{\alpha/2}(x) - y, \ y - \hat{q}_{1-\alpha/2}(x)\right\}$, and use this to make prediction sets. One can verify that the resulting prediction sets are a calibrated version of $\left[\hat{q}_{\alpha/2}, \ \hat{q}_{1-\alpha/2}\right]$ (either expanded or shrunk symmetrically). The intuition is that the CQR score remains small within the learned central quantile band and increases linearly into the tails. For common unimodal distributions, this ordering is approximately monotone with $1 - p(y \mid x)$, so thresholding the CQR score closely emulates the optimal rule. Prediction sets of this form have shown strong performance in terms of average set size in practice. We generalize the idea behind CQR to design a score function tailored to our two-threshold setting. The idea is to learn two distinct sets of quantile functions: one for counterfactual harm when $Y \in H(x)$ and one for complementarity when $Y \notin H(x)$. To achieve this, we learn two pairs of quantile functions, $(\hat{q}_{\varepsilon/2}, \hat{q}_{1-\varepsilon/2})$ for the counterfactual-harm constraint and $(\hat{q}_{\delta/2}, \hat{q}_{1-\delta/2})$ for the complementarity constraint. From these quantile estimates, we define the nonconformity score as $\hat{s}(x, y) :=$

$$
\begin{cases}
\max\{\hat{q}_{\varepsilon/2}(x) - y, \ y - \hat{q}_{1-\varepsilon/2}(x)\}, & y \in H(x), \\
\max\{\hat{q}_{\delta/2}(x) - y, \ y - \hat{q}_{1-\delta/2}(x)\}, & y \notin H(x).
\end{cases}
$$

This score treats labels inside $H(x)$ differently from those outside it, applying a distinct CQR-style score to each in an intuitive manner: for $y \in H(x)$, the score is derived from the counterfactual-harm rate $\varepsilon$; for $y \notin H(x)$, it is derived from the complementarity rate $1 - \delta$.

# 4. Finite Sample Algorithms

We have derived the form of optimal collaborative prediction sets in Theorem 2.1, where we have also discussed strategies for designing the score $s$ in regression and classification. In this section, we fix the conformity score and focus on how to estimate the thresholds $a$ and $b$ from data. We introduce *Collaborative Uncertainty Prediction*-(CUP), an algorithm for constructing collaborative prediction sets in finite samples. We consider two scenarios: (i) the offline setting, where calibration and test data are exchangeable, and (ii) the online setting, where data arrive sequentially and the underlying distribution may drift in unknown ways.

## 4.1. CUP - Offline

In the offline setting, we assume access to a held-out calibration dataset $\mathcal{D}_{\text{cal}} = \{(X_i, Y_i, H(X_i))\}_{i=1}^{n}$ that is exchangeable with the test data $\mathcal{D}_{\text{test}} = \{(X_j, Y_j, H(X_j)))\}_{j=1}^{m}$. The goal is to estimate the thresholds $(\hat{a}, \hat{b})$ and implement the set $C(x) = \{y : \ s(x, y) \leq \hat{a}\,\mathbf{1}\{y \notin H(x)\} + \hat{b}\,\mathbf{1}\{y \in H(x)\}\}$. For each calibration point $(x_i, y_i)$, we compute a non-conformity score $s_i = s(x_i, y_i)$, and separate the scores

into two groups according to whether the true label lies in the human set or not. The thresholds are then obtained by taking empirical quantiles of these two groups:

$$
\hat{b} = \text{Quantile}_{1-\varepsilon}\Big(\{\, s_i : Y_i \in H(X_i)\,\} \cup \{\infty\}\Big),
$$

$$
\hat{a} = \text{Quantile}_{1-\delta}\Big(\{\, s_i : Y_i \notin H(X_i)\,\} \cup \{\infty\}\Big).
$$

The following Proposition shows these sets satisfy finite-sample guarantees.

**Proposition 4.1** (Finite-Sample Offline Guarantees)**.** *Let* $(X_{n+1}, Y_{n+1}, H(X_{n+1}))$ *be a new test point, exchangeable with the calibration data. Let* $n_1$ *be the number of calibration points where* $Y_i \in H(X_i)$ *and* $n_2$ *the number where* $Y_i \notin H(X_i)$. *The thresholds* $\hat{a}$ *and* $\hat{b}$ *satisfy:*

$$
\mathbb{P}(Y_{n+1} \in C(X_{n+1}) \mid Y_{n+1} \in H(X_{n+1})) \ \geq \ 1 - \varepsilon \quad and
$$
$$
\mathbb{P}(Y_{n+1} \in C(X_{n+1}) \mid Y_{n+1} \notin H(X_{n+1})) \ \geq \ 1 - \delta.
$$

*Also, if the scores have continuous distribution, then:*

$$
\mathbb{P}(Y_{n+1} \in C(X_{n+1}) \mid Y_{n+1} \in H(X_{n+1})) < 1 - \varepsilon + \frac{1}{n_1 + 1},
$$

$$
\mathbb{P}(Y_{n+1} \in C(X_{n+1}) \mid Y_{n+1} \notin H(X_{n+1})) < 1 - \delta + \frac{1}{n_2 + 1}.
$$

The assumption of exchangeability for lower bound and continuity for upper bounds are both common in the conformal prediction literature (e.g., (Vovk et al., 2005)).

In practice, exchangeability is fragile, and real-world distribution shifts inevitably undermine the validity of offline guarantees. Shifts may stem from many sources, but in the context of long-term human-AI collaboration, a particularly salient one is what we call *Human-to-AI Adaptation*. As collaboration unfolds, humans may gradually adjust how they construct their sets $H(x)$ in response to the AI's behavior. For instance, the human might learn over time which types of instances–such as which patients in a medical setting–the AI tends to be more knowledgeable about, and tune their proposals accordingly to be maximally helpful to the final set. In some cases, this may mean proposing larger sets to improve coverage, while in others it may mean offering smaller, more decisive sets to sharpen outcomes. Such feedback loops alter the distribution of test-time data and violate exchangeability between calibration and test sets. To address this, we now turn to the *online setting*, which relaxes allows the data distribution to evolve over time.

## 4.2. CUP - Online

In this Section, we move to the online setting where data arrives sequentially, one sample at a time. At each round $t$, the test input $x_t$ and the human's proposed set $H(x_t)$

are provided to the AI, which must then output the final prediction set $C_t(x_t)$. Only after the final prediction set is announced is the true label $y_t$ revealed. Here, we make no assumptions about the distribution of the data stream, an assumption particularly natural for human-AI collaboration, where distribution shift is not merely accidental but may arise directly from the interaction itself.

We design an online algorithm, CUP–Online, that makes prediction sets of the form, $C_t(x_t) = \{y \in \mathcal{Y} \mid s(x_t, y) \leq a_t \mathbf{1}\{y \notin H(x_t)\} + b_t \mathbf{1}\{y \in H(x_t)\}\}$, where $s(.,.)$ is a fixed non-conformity score, and $(a_t, b_t)$ are the two thresholds that we will update online. Let us also define

$$\text{err}_t^{\text{in}} := \mathbf{1}\{y_t \notin C_t(x_t),\ y_t \in H(x_t)\},$$
$$\text{err}_t^{\text{out}} := \mathbf{1}\{y_t \notin C_t(x_t),\ y_t \notin H(x_t)\}.$$

Then, fixing a learning rate $\eta > 0$, CUP–Online updates only one threshold at a time, depending on whether the human included the true label in their proposed set.

$$
\begin{aligned}
\mathbf{if}\ y_t \in H(x_t):\ & b_{t+1} = b_t + \eta\left(\mathbf{1}\{s(x_t, y_t) > b_t\} - \varepsilon\right), \\
& a_{t+1} = a_t \\
\mathbf{if}\ y_t \notin H(x_t):\ & a_{t+1} = a_t + \eta\left(\mathbf{1}\{s(x_t, y_t) > a_t\} - \delta\right), \\
& b_{t+1} = b_t
\end{aligned}
$$

Intuitively, if errors occur more often than expected, the threshold is relaxed to include more labels, if errors are too rare, the threshold is tightened. Over time this feedback process drives the empirical error rates toward their target values $\varepsilon$ and $\delta$. The choice of $\eta$ gives a tradeoff between adaptability and stability, while larger values will make the method more adaptive to observed distribution shifts (this will also show up in our guarantees) they also induce greater volatility in thresholds values, which may be undesirable as it allows the method to fluctuate between smaller sets to larger sets. Hence, in practice, a hyperparameter tuning for $\eta$ can enhance the performance of CUP–Online. We now outline the theoretical guarantees of our online algorithm.

**Proposition 4.2** (Finite-Sample Guarantees). *Assume the conformity is bounded, i.e, $s(x, y) \in [0, 1]$ and let $N_1(T) = \sum_{t=1}^{T} \mathbf{1}\{y_t \in H(x_t)\}$ and $N_2(T) = \sum_{t=1}^{T} \mathbf{1}\{y_t \notin H(x_t)\}$. For any $T \geq 1$:*

$$\left| \frac{1}{N_1(T)} \sum_{t=1}^{T} \text{err}_t^{\text{in}} - \varepsilon \right| \leq \frac{1 + \eta \max(\varepsilon, 1 - \varepsilon)}{\eta N_1(T)},$$

$$\left| \frac{1}{N_2(T)} \sum_{t=1}^{T} \text{err}_t^{\text{out}} - \delta \right| \leq \frac{1 + \eta \max(\delta, 1 - \delta)}{\eta N_2(T)}.$$

*Remark* 4.3. The boundedness assumption on the conformity score holds automatically in classification when $s$ is

derived from probabilistic outputs (e.g., a softmax score, which lies in $[0, 1]$). In regression, this condition can be enforced by rescaling and clipping the score.

The quantities controlled in Proposition 4.2 are the empirical counterparts of the counterfactual harm and complementarity rates defined in (1) and (2). The bounds show that these empirical rates converge to their targets at a rate proportional to $1/N_1(T)$ and $1/N_2(T)$, respectively, hence vanish as the number of relevant interactions grows. Importantly, these guarantees are distribution-free and hold under arbitrary data sequences, covering settings with human-to-AI adaptation and other shifts. Bounds of this form exist in online conformal prediction for controlling marginal error rates (Gibbs & Candès, 2021; Angelopoulos et al., 2023; Ramalingam et al., 2025), and our contribution is to extend them to the simultaneous control of harm and complementarity.

## 5. Experiments

First, we outline our experimental setup, and then evaluate our framework across three distinct data modalities: (i) image classification, (ii) real-valued regression, and (iii) text based medical decision-making with large language models. For each modality, we study both the offline and online algorithms introduced in Section 4. We additionally validate our findings with a real-human crowdsourcing study (Appendix C.1) with results consistent with the findings reported here.

**Baselines.** We compare against the following natural baselines: *(i) Human alone.* which uses the human-proposed set $H(x)$ directly, without any AI refinement. We treat the human policy as a black box and make no assumptions about how the sets are generated. Coverage depends entirely on the provided sets and may vary with human expert quality These human sets are constructed using crowd-sourced annotations, rule-based diagnostic systems or synthetic noise, depending on the task. Full details are provided in each experiment subsection. *(ii) AI alone*, which uses the AI system without incorporating human input, reducing to standard conformal prediction based solely on the model scores. This provides a benchmark for how well the AI performs independently. Additionally, in the online setting we consider a fixed baseline that serves as a reference point for detecting and evaluating distribution shifts. This method uses a static set of thresholds computed from an initial subset of data (i.e early examples or a dedicated split), and then applies these thresholds over the online data stream without any further updates. This baseline provides a useful comparison to understand the value of adaptivity in the online setting.

**Evaluation metrics.** Across all experiments, we evaluate methods based on two key quantities: *marginal coverage*, the probability that the true label lies in the prediction set,

and *average set size*, measured as cardinality in classification and interval length in regression. In the online setting, we use *running* versions of these metrics, defined at each time step $t$ as $\widehat{\text{cov}}_t = \frac{1}{t}\sum_{i=1}^{t}\mathbf{1}\{y_i \in C(x_i)\}$ for marginal coverage and $\widehat{\text{size}}_t = \frac{1}{t}\sum_{j=1}^{t}|C(x_j)|$ for average set size. These metrics capture a central tradeoff: higher coverage is desirable, but must be balanced against set informativeness. Our algorithm does not explicitly enforce a fixed marginal coverage. Instead, the counterfactual harm parameter $\varepsilon$ and the complementarity parameter $\delta$ shape the resulting coverage and set size. By adjusting these parameters, we navigate tradeoffs between the two metrics.

A successful human–AI collaboration should improve upon the human baseline in at least one dimension, coverage or set size, without significantly worsening the other. In the best case, both metrics improve together. The better the AI model, the more effectively it should recover missed outcomes without unnecessarily inflating sets. Similarly, the stronger the human baseline, the better the collaborative procedure can perform, since it starts from a higher-quality initial proposal. Our framework enables us to study how the quality of the final outputs depends on both human and AI combined quality across our experimental tasks.

### 5.1. Classification: ImagetNet-16H

Our first set of experiments use the ImageNet-16H dataset (Steyvers et al., 2022), which captures human prediction behavior under varying perceptual noise. It consists of 32,431 human predictions on 1,200 natural images, each annotated by multiple participants and perturbed with one of four noise levels $\omega \in \{80, 95, 110, 125\}$ that progressively increase task difficulty. The label space is restricted to a fixed set of 16 classes. As AI, we use a pre-trained VGG19 classifier (Simonyan & Zisserman, 2015) fine-tuned for 10 epochs. We evaluate our framework in an offline setting and then in an online setting with various distribution shifts.

**Offline Setting.** We compare three approaches: *Human Alone*, *AI Alone*, and CUP-offline. Results are averaged over 10 random calibration/test splits. Table 1 reports coverage and set size under two representative noise levels, $\omega = 95$ and $\omega = 125$. For the human baseline, we aggregate multiple annotations into empirical label frequencies and form top-$k$ sets by selecting the $k$ most frequently chosen labels. From the algorithm's perspective, only the sets—not raw annotations or confidences—are observed. The AI baseline applies standard conformal prediction without human input. Since conformal methods allow direct control over target coverage, we evaluate AI Alone at the same realized coverage achieved by CUP-offline. This ensures a fair comparison, where the only dimension for improvement is set size (i.e., if CUP achieves the same coverage with smaller sets, it shows that human input is being used effectively to

tighten predictions). CUP-offline incorporates both sources, with coverage and size determined by $(\varepsilon, \delta)$ parameters that encode counterfactual harm and complementarity.

We include two noise levels to evaluate performance under varying task difficulty for the human. As shown in Table 1, across both levels, our CUP-offline consistently improves on the human baseline. When the human sets are relatively large(e.g top -2), CUP-offline yields strict improvements in both dimensions, reducing set size while improving coverage. At $\omega = 125$, for example, human top-2 sets cover 80% of labels with size 2.0, whereas CUP-offline improves coverage to 90% while reducing size to 1.49. When human sets are very small (e.g., top-1), coverage improvements typically requires adding labels, slightly increasing set size. Even then, CUP-offline offers more efficient sets than AI Alone, leveraging human input to achieve better tradeoffs. At $\omega = 95$, for example, CUP-offline achieves 97.6% coverage with an average size of 1.43, whereas AI Alone requires size 2.27 for similar coverage. Overall, CUP-offline improves on raw human sets and produces tighter sets than AI Alone, adapting to the strengths of each source to provide a clear advantage over both baselines.

**Online Setting.** We consider two types of shifts: a *noise shift*, where inputs are ordered from high to low noise levels ($\omega = 125 \rightarrow 95$), and a *human strategy shift*, where human sets evolve from top-2 to top-3 strategies. The latter serves as an instance of what we term *Human-to-AI Adaptation* which in this case is how humans might adapt their behavior in response to increasing task difficulty or AI feedback.

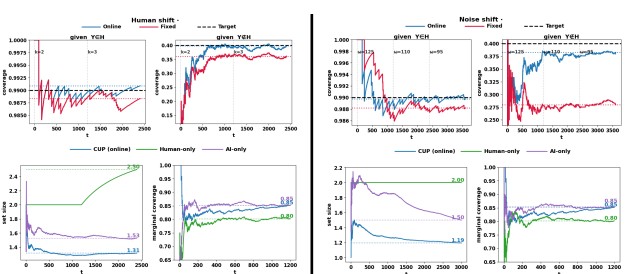

*Figure 2.* **ImageNet-16H- Online Results:** Performance under human strategy shift (left) and noise shift (right). Top: running coverage for CUP-online vs fixed baseline. Bottom: CUP-online vs human-only and AI-only baselines on running set size and marginal coverage.

We first compare CUP-online to the fixed baseline tuned on a separate segment of the data stream. For instance, in the noise shift setting, we tune $(a, b)$ on $\omega = 80$, and for the human shift, on top-1 human prediction sets. To evaluate, we track *constraint-specific coverage* over time. At each time step $t$, we compute $\text{cov}_t^* = 1 - 1/t \sum_{i=1}^{t} 1 - \text{err}_i^*$, where $\text{err}_i^*$ is either a counterfactual harm error or a complementarity error defined in Section 4.2. Intuitively, this metric tracks how well the algorithm maintains the

| | $\omega = 125$ | | | | | | | |
|---|---|---|---|---|---|---|---|---|
| | **Human Alone** | | **CUP** | | | | **AI Alone** | |
| Strategy | Coverage | Size | Coverage | Size | $\varepsilon$ | $\delta$ | Coverage | Size |
| Top-2 | $0.8008 \pm 0.0090$ | $2.00 \pm 0.00$ | $\mathbf{0.9022 \pm 0.0083}$ | $\mathbf{1.49 \pm 0.04}$ | 0.05 | 0.70 | $0.9072 \pm 0.0138$ | $1.65 \pm 0.07$ |
| Top-1 | $0.7245 \pm 0.0103$ | $1.00 \pm 0.00$ | $\mathbf{0.8823 \pm 0.0134}$ | $1.36 \pm 0.07$ | 0.05 | 0.70 | $0.8828 \pm 0.0140$ | $1.48 \pm 0.05$ |

| | $\omega = 95$ | | | | | | | |
|---|---|---|---|---|---|---|---|---|
| | **Human Alone** | | **CUP** | | | | **AI Alone** | |
| Strategy | Coverage | Size | Coverage | Size | $\varepsilon$ | $\delta$ | Coverage | Size |
| Top-2 | $0.9613 \pm 0.0061$ | $2.00 \pm 0.00$ | $\mathbf{0.9825 \pm 0.0066}$ | $\mathbf{1.77 \pm 0.44}$ | 0.01 | 0.80 | $0.9830 \pm 0.0061$ | $2.10 \pm 0.15$ |
| Top-1 | $0.9257 \pm 0.0060$ | $1.00 \pm 0.00$ | $\mathbf{0.9763 \pm 0.0076}$ | $1.43 \pm 0.07$ | 0.01 | 0.80 | $0.9755 \pm 0.0053$ | $2.27 \pm 0.21$ |

*Table 1.* **ImageNet-16H – Offline Results:** Comparison of Human, AI, and CUP under two noise levels. Reports marginal coverage and average set size (mean ± std over 10 splits). CUP uses calibration parameters $(\varepsilon, \delta)$.

target coverage level over time.

Figure 2 (top row) shows the results for both forms of distribution shift: human strategy shift (left) and noise shift (right). In both cases, the online algorithm remains close to the target coverage levels throughout the stream, while the fixed baseline drifts away and fails to recover from the changes in the underlying distribution. In the bottom row of Figure 2, we compare CUP-online with human-only and AI-only baselines, using the running marginal coverage and set size metrics defined earlier. For a fair comparison, we run the AI-only baseline at a target coverage level matched to the realized coverage achieved by CUP-online across the full stream. The results show that CUP-online consistently improves over the human baseline by achieving higher coverage while keeping the sets small. Compared to AI alone, where coverage is matched by design, CUP-online produces more compact sets. These trends mirror those seen in the offline setting, showing we maintain the advantages of the collaborative framework under distribution shifts.

### 5.2. LLMs for Medical Diagnosis Decision Making

Our second set of experiments evaluates the framework in the text modality of data, focusing on a medical decision-making task using the DDXPlus dataset (Fansi Tchango et al., 2022). This dataset contains synthetic patient records generated from a medical knowledge base and rule-based diagnostic system. Each record includes demographics, symptoms, and antecedents linked to an underlying condition, along with a differential diagnosis list. From this list we form human prediction sets using a top-$k$ strategy, where the human provides the $k$ most likely diagnoses. For the AI component, we use two language models with contrasting accuracy: **GPT-5**, which performs strongly, and **GPT-4o**, which is weaker and often falls below the human baseline. This contrast highlights how the quality of the AI model shapes the trade-offs of collaboration.

**Offline Setting.** Tables 2 summarize the results for GPT-4o

and GPT-5, respectively, under two different human strategies. Across all settings, CUP-offline improves on the human baseline by raising coverage, as the procedure explicitly augments human sets when the true label is missing. Naturally, this may increase set size, but when the AI is sufficiently strong, as with GPT-5, the algorithm is able to both prune away incorrect human labels and add the correct label when necessary more efficiently. This yields prediction sets that improve across both dimensions: achieving higher coverage *and* smaller size, outperforming both baselines.

With a weaker model such as GPT-4o, coverage gains may come at the cost of slightly larger sets, reflecting that the model's weaker capability This does not undermine the approach but rather illustrates the role of the AI component in determining the ultimate efficiency of the collaborative sets. Still, CUP-offline produces smaller sets than AI alone at comparable coverage levels, showing that human knowledge is being used productively. Taken together, these results demonstrate that CUP-offline yields consistent benefits over both the human and AI baselines, while the degree to which coverage and set size can be simultaneously optimized depends on the strength of the AI model.

**Online Setting.** We next evaluate the CUP-online algorithm in the medical setting, using GPT-5 as the AI. Human sets follow a top-$k = 2$ strategy, and distribution shift is induced by ordering test patients by age, from younger to older groups. We also include a non-adaptive baseline with fixed thresholds tuned on the earliest segment (ages 1–30) to isolate the effect of adaptivity in the face of demographic change, however due to space constraints direct comparison with this baseline is deferred to the Appendix C.3.

As in the ImageNet experiments, we benchmark CUP-online against human-only and AI-only baselines. Figure 3 shows results under demographic shift. The pattern is consistent with the offline setting: CUP improves over the human baseline in both coverage and set size, and against AI Alone it achieves smaller sets at matched coverage.

| | Human | GPT-4o | | | GPT-5 | | |
|---|---|---|---|---|---|---|---|
| Strategy | C/S | CUP C/S | $(\varepsilon, \delta)$ | AI C/S | CUP C/S | $(\varepsilon, \delta)$ | AI C/S |
| Top-1 | 0.71 / 1.00 | 0.90 / 2.84 | (0.02, 0.70) | 0.88 / 4.64 | 0.91 / 1.59 | (0.02, 0.70) | 0.91 / 1.76 |
| Top-2 | 0.87 / 1.95 | 0.93 / 3.14 | (0.01, 0.45) | 0.90 / 9.12 | 0.93 / 1.65 | (0.02, 0.45) | 0.93 / 1.95 |

*Table 2.* **LLMs-Offline Results** Entries report coverage/size (C/S). Calibration parameters $(\varepsilon, \delta)$ shown for CUP.

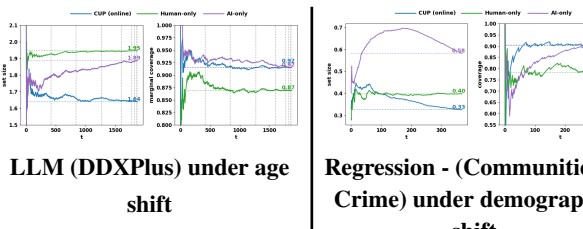

| LLM (DDXPlus) under age shift | Regression - (Communities & Crime) under demographic shift |
|---|---|

*Figure 3.* Online results: LLM (left) and regression (right), comparing CUP–online with baselines Human and AI.

| Human A | | | |
|---|---|---|---|
| Human A C/S | CUP C/S | $(\varepsilon, 1-\delta)$ | AI C/S |
| 0.760 / 0.581 | 0.862 / 0.380 | (0.10, 0.70) | 0.862 / 0.394 |
| 0.760 / 0.581 | 0.825 / 0.326 | (0.15, 0.70) | 0.825 / 0.337 |
| **Human B** | | | |
| Human B C/S | CUP C/S | $(\varepsilon, 1-\delta)$ | AI C/S |
| 0.872 / 0.618 | 0.948 / 0.528 | (0.05, 0.90) | 0.948 / 0.608 |
| 0.872 / 0.618 | 0.953 / 0.558 | (0.05, 0.95) | 0.953 / 0.588 |

*Table 3.* Regression–Offline Results: Coverage/size (C/S) under two human expert settings.

## 5.3. Regression: Communities & Crime

Finally, we look at a regression task using the UCI *Communities & Crime* dataset (Redmond, 2002), where the goal is to predict the violent crime rate per community. To simulate human input, we generate intervals centered around noisy point estimates of the ground truth. Specifically, we perturb each true label with Gaussian noise to form $\hat{y}(x)$, then construct the interval $H(x) = [\hat{y}(x) - w(x)/2, \hat{y}(x) + w(x)/2]$, where $w(x)$ is a base width also subject to noise. By varying the noise levels, we simulate human experts of differing quality. As in earlier experiments, the algorithm only observes the final set $H(x)$, not how it was generated. The AI model builds on the setting explained in Section 3. We train two MLPs using the pinball loss to estimate conditional quantiles: one for predicting $(\hat{q}_{\varepsilon/2}, \hat{q}_{1-\varepsilon/2})$, and one for $(\hat{q}_{\delta/2}, \hat{q}_{1-\delta/2})$. Each model shares a backbone with two output heads. The resulting four quantiles define the CQR-style score in Section 3, to which we apply the CUP-offline procedure to obtain the two thresholds used at test time.

We compare *Human Alone*, which uses the raw intervals $H(x)$; *AI Alone*, which applies standard conformalized quantile regression without access to the human sets; and CUP-offline, which combines both sources via the proposed collaborative algorithm. Results are reported in Table 3 for two human experts of different quality.

First, we note that CUP improves upon the human baseline in terms of both coverage and interval width. Second, the results highlight the complementary role of human, with greater gains observed over AI Alone when initial human input is of higher quality. This complements the medical diagnosis results, where we varied the AI instead of the human. Together, the two experiments show that the collaboration efficiency depends on the quality of both parties.

**Online Setting.** We evaluate CUP-online under a controlled distribution shift based on community demographics. Test examples are ordered by the proportion of residents identified by a randomly selected race-coded variable. Figure 3 reports the running marginal coverage and average set size for AI, Human, and CUP-online. Consistent with the previous experiments, we again observe that CUP-online improves upon both baselines: compared to Human Alone, it increases coverage without inflating intervals; compared to AI Alone, it reduces interval width while preserving coverage. The non-adaptive baseline with fixed thresholds is tuned on the earliest portion of the stream (i.e., communities with the lowest demographic proportion). Results for this baseline are deferred to Appendix 9. All together, the results underscore the robustness of the collaborative approach across modalities and under shifting data distributions.

## 6. Conclusion

We introduced a framework for constructing prediction sets collaboratively between humans and AI, grounded in two core principles: avoiding counterfactual harm and enabling complementarity. We showed that the optimal sets take a simple two-threshold form, and developed finite-sample algorithms for both offline and online settings. Across diverse domains and agents (human or AI) strengths, our methods consistently leverage the collaboration capabilities of human and AI to produce sets that outperform either alone. This framework offers a principled and practical approach to structured collaboration under uncertainty.

## Acknowledgements

The authors thank EnCORE, the Institute for Emerging CORE Methods in Data Science, for their support. SK additionally acknowledges support from a gift from AWS to Penn Engineering's ASSET Center for Trustworthy AI.

## Impact Statement

This work provides a principled framework for human–AI collaborative uncertainty quantification, characterizing optimal ways for an AI to prune and augment a human's prediction set through a two-threshold rule and giving offline/online calibration algorithms with distribution-free guarantees. It enables safer collaboration by explicitly controlling counterfactual harm while ensuring complementarity, even under distribution shift and evolving human behavior.

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

# A. Extended Related Works

**Conformal Prediction** The idea of constructing prediction regions can be traced back to classical work on tolerance intervals in statistics (Wilks, 1941; Scheffe & Tukey, 1945). Modern Conformal Prediction (CP), introduced by (Vovk et al., 1999; Saunders et al., 1999; Vovk et al., 2005), builds on this foundation to provide distribution-free, finite-sample validity: given a desired confidence level, CP guarantees that the constructed prediction set contains the true outcome with the prescribed marginal probability.

Over the past two decades, CP has become a standard tool in machine learning for both classification and regression tasks (Papadopoulos et al., 2002; Lei et al., 2017; Romano et al., 2019; 2020), and recently extended to language models (Noorani et al., 2025a; Quach et al., 2024; Mohri & Hashimoto, 2024; Cherian et al., 2024; Su et al., 2024) with a large literature on improving *efficiency* (shrinking set size while preserving coverage) (Fisch et al., 2024; Gupta et al., 2022; Kiyani et al., 2024; Stutz et al., 2022; Noorani et al., 2025b). A growing body of work extends CP beyond marginal coverage to control more general notions of risk. (Angelopoulos et al., 2025) introduced conformal risk control, showing how prediction sets can be calibrated to satisfy monotone risk measures rather than coverage alone. (Lindemann et al., 2023) applied these principles to safe planning in dynamic environments, demonstrating how conformal methods can enforce operational safety constraints. (Lekeufack et al., 2024) developed a conformal decision-theoretic framework where decisions are parameterized by a single scalar and calibrated to control risk. (Cortes-Gomez et al., 2025) expands on this view by developing utility-directed conformal prediction, which constructs sets that both retain standard coverage guarantees and minimize downstream decision costs specified by a user-defined utility function. More broadly, (Kiyani et al., 2025) show that prediction sets can be viewed as a natural primitive for risk-sensitive decision making: they communicate calibrated uncertainty in a form well-suited for risk-averse decision makers operating in high-stakes domains. This perspective makes conformal prediction sets relevant for human–AI collaboration, particularly in high-stakes applications, where reliable uncertainty estimates are essential for enabling trust and complementarity between human expertise and machine predictions. Our work provides a principled way to generalize the methodologies developed by these studies to scenarios where humans are in the loop, contributing to the final design of prediction sets.

**Human-AI collaboration** Human–AI decision-making has attracted growing interest across the machine learning community and social sciences (Guo et al., 2024; Marusich et al., 2024; Guo et al., 2025; Hullman et al., 2019). Yet, realizing true complementarity where the joint system outperforms either the human or the AI-alone, or both, remains challenging (Lai et al., 2021; Vaccaro et al., 2024; Bansal et al., 2021). Interestingly, a recent meta analysis by (Vaccaro et al., 2024) found out that, on average, human–AI teams under-perform the stronger individual agent. These findings underscore persistent difficulties around coordination, trust, and communication between machine and human, motivating the need for algorithmic frameworks that can systematically structure collaboration. From a theoretical perspective, (Donahue et al., 2024) characterize the conditions under which providing two sets of recommendations outperforms a single set, highlighting that the benefits of joint decision-making depend heavily on the diversity of errors and the specific aggregation rule used. This work takes a step toward a mathematically rigorous framework that combines the power of AI and humans in a way that complements human capabilities while controlling potential counterfactual harm.

**Learning to Defer**. One approach is the *learning to defer* (L2D) paradigm, where the AI model learns when to predict on its own and when to defer to a human expert. Earlier work (Madras et al., 2018) framed this as a mixture-of-experts problem, jointly training a classifier with a deferral mechanism. (Wilder et al., 2021) extended this with a decision-theoretic formulation, training models to complement human strengths rather than maximize accuracy alone. Subsequent work studied the design of surrogate losses for deferral, for example (Mozannar & Sontag, 2021) showed that standard training objectives can fail to produce optimal deferral policies and proposed a consistent surrogate loss that guarantees Bayes-optimal deferral. Extensions address various settings: (Verma & Nalisnick, 2022) and (Charusaie et al., 2022) studied deferral with multiple experts, while (Wei et al., 2024) emphasized that humans and models are not independent and introduced dependent Bayes optimality to exploit correlations between them. (Okati et al., 2021) formulated differentiable learning under triage, providing exact optimality guarantees for multi-expert deferral. (Bary et al., 2025) proposed a training-free deferral framework that leverages conformal prediction to allocate decisions among multiple experts. And most recently, along these ideas, (Arnaiz-Rodriguez et al., 2025) introduced a collaborative matching system that selectively defers to humans to maximize overall performance.

Overall, the L2D literature focuses on *who decides* on each instance: the model or the human. These methods improve team performance by abstention or delegation, which is inherently different than our approach. We start from the human's proposed *set* and ask how to refine it with AI. The goal is to always produce a combined prediction set, particularly one that

is simultaneously more reliable and more informative than either agent alone. In this sense, our approach complements deferral-based methods but addresses a different question: not *who decides*, but *how to decide together*.

**Agreement protocols**

Another line of work views collaboration as an interactive process through *agreement protocols*, where humans and models iteratively exchange feedback until consensus is reached (Aumann, 1976; Collina et al., 2025b; Geanakoplos & Polemarchakis, 1982; Aaronson, 2004; Collina et al., 2024). Earlier formulations (Aaronson, 2004) assume perfect Bayesian updates under a common prior, and make assumptions that generally preclude tractable implementation. (Collina et al., 2024) generalize these results and introduce tractable agreement protocols that replace Bayesian rationality with statistically efficient calibration conditions which enable agreement theorems without distributional assumptions. More recent generalizations (Collina et al., 2025a) develop collaboration protocols that achieve information aggregation as well in the setting where agents observe different features of the same instance. These approaches assume that both agents maintain probabilistic beliefs and communicate probability vectors or expected-value estimates until their predictions become sufficiently close. In contrast, our framework differs in scope and considers a setting where the human provides only a prediction set, without the need to specify probabilities or beliefs over labels. Moreover, while agreement protocols emphasize belief convergence between agents, our work prioritizes the human and focuses on constructing collaborative prediction sets that balance the two objectives of counterfactual harm and promoting complementarity. The agreement protocol literature therefore captures a complementary aspect of human–AI collaboration, centering on belief alignment rather than prediction-set construction, and it remains nontrivial to translate the converged beliefs in these protocols into prediction sets that satisfy counterfactual harm or complementarity criteria.

**Prediction sets for human-AI decision support** A more related and recent strand of work has explored prediction sets as a structured interface for collaboration and human decision making support. (Straitouri et al., 2023) formalized the problem of improving expert predictions with Conformal Prediction in multiclass classification. In their setting, the AI provides a subset of candidate labels for each instance, from which the human selects, ensuring that the advice is structured but does not override the expert's agency. Similarly, (Benz & Rodriguez, 2024) argue that standard calibration is insufficient for human-AI teams because it ignores the human's existing knowledge. They propose human-aligned calibration, which ensures that AI-generated prediction sets are calibrated specifically on the instances where the human is likely to be incorrect or uncertain, thereby maximizing the utility of the AI's intervention. In parallel, (Babbar et al., 2022) empirically evaluated prediction sets in human–AI teams, and showed that set-valued advice can improve human accuracy compared to single-label predictions. However, they also found that large prediction sets may confuse or slow down human decision-making. To mitigate this, they introduced Deferral-CP (D-CP), where the AI is allowed to abstain entirely on instances for which no sufficiently small set can be produced, deferring the decision back to the human. Closely related, (Hullman et al., 2025) analyzed Conformal Prediction from a decision-theoretic perspective, examining how coverage guarantees relate to human goals and strategies in decision making. They formalize possible ways a decision maker might use a prediction set and highlight tensions between conformal coverage and the forms of uncertainty information that best support human decisions. Other works have studied how to design prediction sets specifically tailored for human use, for example (De Toni et al., 2024) proposed a greedy algorithm for constructing prediction sets and showed empirically that it can improve average human accuracy compared to standard conformal sets.

While these studies focus on how prediction sets can support human decision-making, i.e. how humans utilize AI-generated sets to make final judgments, our framework addresses a complementary question: how human feedback can be used to construct a more reliable collaborative prediction set. Rather than treating the human as the end decision-maker, we model the prediction set itself as the outcomes of collaboration, designed jointly to reflect human and AI strengths.

**Counterfactual harm and complementarity** In recent years, there has been growing concern about the unintended consequences of decision support systems using machine learning algorithms in high-stakes domains (Richens et al., 2022; Li et al., 2023; Beckers et al., 2022). To this end, (Straitouri et al., 2024) analyze decision-support systems based on prediction sets through the lens of *counterfactual harm* (Feinberg, 1986).Their concern is that requiring humans to always select from a machine-provided set may, in some cases, harm performance: a human who would have been correct unaided might be misled by the system. Using structural causal models, they formally defined and quantified this notion of harm, and under natural monotonicity assumptions, provided methods to estimate or bound how frequently harm may occur without deploying the system. While closely related to our framework, their setting differs from ours in that they study systems where the AI supplies sets from scratch, and their definition of counterfactual harm focuses on the subsequent prediction accuracy of the human, whereas we start from sets already provided by the human and ask how to refine them collaboratively,

and our definition of counterfactual harm is a direct measure of the quality of the refining procedure.

On the other hand, in the broader human-AI collaboration literature, complementarity is typically defined as whether the combined system achieves higher average accuracy than either the human or the model alone (Yin et al., 2019; Suresh et al., 2020; Lai et al., 2021; Benz & Rodriguez, 2025), and it is currently still unclear how to guarantee this. Our formulation of complementarity is different: it is set-based rather than accuracy-based. Instead of asking whether joint predictions improve overall accuracy, we require that the collaborative prediction set recovers outcomes the human initially missed, while simultaneously avoiding counterfactual harm. This shifts the focus from point-prediction accuracy to the tradeoff between set-based coverage and set size. Crucially, by defining complementarity in this way, our framework provides a principled way to formalize and guarantee it, with clear tradeoffs between the two central metrics of uncertainty quantification.

# B. Proofs

## B.1. Proof of Theorem 2.1 ( Optimal Prediction Sets )

*Proof.* The primary optimization problem is given by:

$$
\begin{aligned}
\min_{C:\mathcal{X}\to 2^{\mathcal{Y}}} \quad & \mathbb{E}\,|C(X)| \\
\text{s.t.} \quad & \mathbb{P}(Y \in C(X) \mid Y \in H(X)) \ge 1 - \varepsilon, \\
& \mathbb{P}(Y \in C(X) \mid Y \notin H(X)) \ge \delta.
\end{aligned}
\tag{P}
$$

**Part 1** (LP Relaxation). *The original problem involves optimizing over a space of discrete sets, which is a combinatorial and generally NP-hard problem. To make it more tractable, we can formulate an equivalent problem using a continuous relaxation. Let $C(x, y) \in [0, 1]$ be a variable indicating the degree to which $y$ is included in the set for instance $x$. With this relaxation, the objective function, which is the expected size of the set, can be re-written as $\mathbb{E}\left[\int_{\mathcal{Y}} C(X, y)dy\right]$. Next, we can use the definition of conditional probability, i.e $P(A|B) = \frac{\mathbb{E}[\mathbf{1}_A \mathbf{1}_B]}{\mathbb{E}[\mathbf{1}_B]}$ to rewrite the constraints in terms of expectations, which leads to the following:*

$$
\begin{aligned}
\min_{C:\mathcal{X}\times\mathcal{Y}\to[0,1]} \quad & \mathbb{E}\left[\int_{\mathcal{Y}} C(X, y)dy\right] \\
\text{s.t.} \quad & \mathbb{E}[C(X, Y)\mathbf{1}_{Y\in H(X)}] \ge (1-\varepsilon)\mathbb{E}[\mathbf{1}_{Y\in H(X)}], \\
& \mathbb{E}[C(X, Y)\mathbf{1}_{Y\notin H(X)}] \ge \delta\mathbb{E}[\mathbf{1}_{Y\notin H(X)}].
\end{aligned}
\tag{$\mathcal{P}_{rel}$}
$$

*Note that $\mathbb{E}_{X,Y}[\mathbf{1}_{Y\in H(X)}] = P[Y \in H(X)]$. This problem is a linear program, where both the objective functions and the constraints are linear with respect to the decision variable $C$. Since the objective is linear ( and thus convex ) and the feasible region is a convex set, this is a convex optimization problem. Therefore strong duality holds (see Theorem 1 Section 8.3 of (Luenberger, 1969)).*

**Part 2** (Minimax Formulation). *We formulate the Lagrangian for the relaxed problem by introducing Lagrange multipliers $\lambda_1, \lambda_2 \ge 0$ for the two constraints:*

$$
\begin{aligned}
g(\lambda_1, \lambda_2, C) =& \mathbb{E}_X\left[\int_{\mathcal{Y}} C(X, y)dy\right] - \lambda_1\left(\mathbb{E}_{X,Y}[C(X, Y)\mathbf{1}_{Y\in H(X)}] - (1-\varepsilon)\mathbb{E}_{X,Y}[\mathbf{1}_{Y\in H(X)}]\right) \\
& - \lambda_2\left(\mathbb{E}_{X,Y}[C(X, Y)\mathbf{1}_{Y\notin H(X)}] - \delta\mathbb{E}_{X,Y}[\mathbf{1}_{Y\notin H(X)}]\right)
\end{aligned}
$$

*The minimax problem is:*

$$
\min_{C} \max_{\lambda_1, \lambda_2 \ge 0} g(\lambda_1, \lambda_2, C)
$$

*By strong duality, we can swap the order of the maximization and minimization:*

$$
\max_{\lambda_1, \lambda_2 \ge 0} \min_{C} g(\lambda_1, \lambda_2, C)
$$

*To perform the inner minimization over $C$, lets first rewrite the Lagrangian in integral form over the joint probability distribution $p(x, y)$.*

$$g = \int_{\mathcal{X}} \int_{\mathcal{Y}} C(x,y) \left[ p(x) - \lambda_1 \mathbf{1}_{y \in H(x)} p(x,y) - \lambda_2 \mathbf{1}_{y \notin H(x)} p(x,y) \right] dxdy + constant.$$

*To minimize this integral, we can minimize the integrand for each point $(x,y)$ independently, and thus the inner minimization over $C(x,y) \in [0,1]$ is pointwise. By using the relationship $p(x,y) = p(x)p(y|x)$ and factoring out $p(x)$, the term multiplying $C(x,y)$ becomes:*

$$p(x) \left[ 1 - \lambda_1 \mathbf{1}_{y \in H(x)} p(y|x) - \lambda_2 \mathbf{1}_{y \notin H(x)} p(y|x) \right]$$

*Since $p(x) \geq 0$, the choice of $C(x,y) \in [0,1]$ that minimizes the expression depends on the sign of the term in brackets. The minimum is attained at the boundaries by setting $C(x,y) = 1$ if the term is negative and $C(x,y) = 0$ if it's positive. This results in an optimal solution $C^*(x,y)$ that is naturally binary-valued*

$$C^*(x,y) = \mathbf{1} \left\{ 1 - \lambda_1 \mathbf{1}_{y \in H(x)} p(y|x) - \lambda_2 \mathbf{1}_{y \notin H(x)} p(y|x) \leq 0 \right\}$$

*and thus the continuous relaxation is tight, as the optimal solution to the relaxed problem is guarantees to be a valid solution for the original problem over the discrete set where $C(x,y) \in \{0,1\}$.*

**Part 3** (Deriving the Final Form). *The condition for including $y$ in the set $C^*(x)$ can be rewritten as:*

$$1 \leq \lambda_1 \mathbf{1}_{y \in H(x)} p(y|x) + \lambda_2 \mathbf{1}_{y \notin H(x)} p(y|x)$$

*This inequality can be simplified by considering the two mutually exclusive cases for any outcome $y \in \mathcal{Y}$:*

- *If $y \in H(x)$, the condition is $1 \leq \lambda_1 p(y|x) \iff p(y|x) \geq 1/\lambda_1$.*

- *If $y \notin H(x)$, the condition is $1 \leq \lambda_2 p(y|x) \iff p(y|x) \geq 1/\lambda_2$.*

*Combining these two conditions, we can express the optimal set $C^*(x)$ as a single thresholding rule on the conditional probability $p(y|x)$. Let $\lambda_1^*$ and $\lambda_2^*$ be the optimal Lagrange multipliers. We define the optimal thresholds as $b^* = 1 - 1/\lambda_1^*$ and $a^* = 1 - 1/\lambda_2^*$. The condition for including $y$ in the optimal set becomes:*

$$1 - p(y|x) \leq a^* \cdot \mathbf{1}\{y \notin H(x)\} + b^* \cdot \mathbf{1}\{y \in H(x)\}$$

*The optimal set can then be written compactly as a single thresholding rule on the score:*

$$C^*(x) = \left\{ y \in \mathcal{Y} \mid s(y|x) \leq a^* \cdot \mathbf{1}\{y \notin H(x)\} + b^* \cdot \mathbf{1}\{y \in H(x)\} \right\}$$

*Since the optimal solution to the relaxed problem is binary and takes this form, it is also the optimal solution to the original problem.*

*This completes the proof.* $\square$

## B.2. Proof of Proposition 4.1 ( Offline Algorithm - Coverage Validity )

*Proof.* Given the calibration set $D_{cal} = \{(X_i, Y_i)\}_{i=1}^n$, let $H : \mathcal{X} \to 2^{\mathcal{Y}}$ be the human set map and $s : \mathcal{X} \times \mathcal{Y} \to \mathbb{R}$ a non-conformity score function. Assume without loss of generality that the (conditional) distribution of the scores is continuous without ties, however in practice this condition is not important as we can always add a vanishing amount of noise to the scores.

For each point define $s_i := S(X_i, Y_i)$ and $h_i = \mathbf{1}\{Y_i \in H(X_i)\}$. Split the indices of the calibration set into two disjoint groups $\mathcal{D}_1 = \{i \leq n : h_i = 1\}$ with size $n_1$ and $\mathcal{D}_2 = \{j \leq n : h_j = 0\}$ with size $n_2$. Given a calibration set $\mathcal{D}_{cal} = \{(X_i, Y_i)\}_{i=1}^n$, let $H : \mathcal{X} \to 2^{\mathcal{Y}}$ denote the human set map, and $s : \mathcal{X} \times \mathcal{Y} \to \mathbb{R}$ a nonconformity score function. For each example, define the conformity score $s_i := s(X_i, Y_i)$ and let $h_i := \mathbf{1}\{Y_i \in H(X_i)\}$ indicate whether the true label was covered by the human set.

A new test point $(X_{test}, Y_{test})$ is assumed to be exchangeable with the full calibration set. This overall exchangeability of the full set of points implied that, conditioned on the event $h_{test} = 1$, the test point is exchangeable with the set of points in

$\mathcal{D}_1$. Similarly, conditioned on the event $h_{test} = 0$, the test point is exchangeable with the set of points in $\mathcal{D}_2$. The prediction set is then defined as

$$C(X_{test}) = \big\{\, y : \; s(X_{test}, y) \;\leq\; \hat{a}\,\mathbf{1}\{y \notin H(X_{test})\} \;+\; \hat{b}\,\mathbf{1}\{y \in H(X_{test})\}\,\big\},$$

$$\text{where} \quad \begin{cases} \hat{a} := \text{Quantile}_{1-\varepsilon}\big(\{s_i : i \in \mathcal{D}_1\} \cup \{\infty\}\big), \\ \hat{b} := \text{Quantile}_{1-\delta}\big(\{s_i : i \in \mathcal{D}_2\} \cup \{\infty\}\big). \end{cases}$$

We now derive a chain of equalities and inequalities for the case of $Y_{test} \in H(X_{test})$:

$$\Pr[Y_{\text{test}} \in C(X_{\text{test}}) \mid h_{\text{test}} = 1] \stackrel{(a)}{=} \Pr[s_{\text{test}} \leq \hat{a} \mid h_{\text{test}} = 1]$$

$$= \Pr\Big[s_{\text{test}} \leq \text{Quantile}_{1-\varepsilon}\big(\{s_i : i \in \mathcal{D}_{\text{in}}\} \cup \{\infty\}\big) \;\Big|\; h_{\text{test}} = 1\Big]$$

$$\stackrel{(b)}{=} \mathbb{E}\Bigg[\frac{1}{n_1 + 1}\sum_{i \in \mathcal{D}_1 \cup \{\text{test}\}} \mathbf{1}\Big\{s_i \leq \text{Quantile}_{1-\varepsilon}\big(\{s_j : j \in \mathcal{D}_1\} \cup \{s_{\text{test}}\}\big)\Big\} \;\Bigg|\; h_{\text{test}} = 1\Bigg]$$

$$\stackrel{(c)}{\geq} 1 - \varepsilon.$$

and analogously the case if $Y_{test} \notin H(X_{test})$:

$$\Pr[Y_{\text{test}} \in C(X_{\text{test}}) \mid h_{\text{test}} = 0] \stackrel{(a)}{=} \Pr\Big[s_{\text{test}} \leq \hat{b} \;\Big|\; h_{\text{test}} = 0\Big]$$

$$= \Pr\Big[s_{\text{test}} \leq \text{Quantile}_{1-\delta}\big(\{s_i : i \in \mathcal{D}_2\} \cup \{\infty\}\big) \;\Big|\; h_{\text{test}} = 0\Big]$$

$$\stackrel{(b)}{=} \mathbb{E}\Bigg[\frac{1}{n_2 + 1}\sum_{i \in \mathcal{D}_2 \cup \{\text{test}\}} \mathbf{1}\Big\{s_i \leq \text{Quantile}_{1-\delta}\big(\{s_j : j \in \mathcal{D}_2\} \cup \{s_{\text{test}}\}\big)\Big\} \;\Bigg|\; h_{\text{test}} = 0\Bigg]$$

$$\stackrel{(c)}{\geq} 1 - \delta.$$

**where**

(a) By definition of $C(\cdot)$ and the thresholds $\hat{a}$ and $\hat{b}$ when: $h_{\text{test}} = 1$ (resp. 0), inclusion is the event $s_{\text{test}} \leq \hat{a}$ (resp. $s_{\text{test}} \leq \hat{b}$).

(b) By exchangeability within the corresponding group: conditional on $h_{\text{test}}$, the set of scores $\{s_i : i \in \mathcal{D}_1\} \cup \{s_{\text{test}}\}$ (or $\mathcal{D}_2$) is exchangeable, so we average the indicator over the $n_{\text{group}} + 1$ equally likely ranks.

(c) By the definition of the empirical quantile: at least a $1 - \varepsilon$ (resp. $\delta$) fraction of the $n_{\text{group}} + 1$ values are $\leq$ that quantile.

Therefor thus far we have established that

$$\Pr[Y_{\text{test}} \in C(X_{\text{test}}) \mid Y_{\text{test}} \in H(X_{\text{test}})] \;\geq\; 1 - \varepsilon, \qquad \Pr[Y_{\text{test}} \in C(X_{\text{test}}) \mid Y_{\text{test}} \notin H(X_{\text{test}})] \;\geq\; 1 - \delta.$$

And now for the upper bounds: for the case $Y_{test} \in H(X_{test})$:

$$\Pr[Y_{\text{test}} \in C(X_{\text{test}}) \mid Y_{\text{test}} \in H(X_{\text{test}})] \stackrel{(d)}{=} \frac{\lceil (1-\varepsilon)(n_1 + 1)\rceil}{n_1 + 1} \stackrel{(e)}{<} 1 - \varepsilon + \frac{1}{n_1 + 1}.$$

Similarly, for the case $Y_{test} \notin H(X_{test})$, we have:

$$\Pr[Y_{\text{test}} \in C(X_{\text{test}}) \mid Y_{\text{test}} \notin H(X_{\text{test}})] = \frac{\lceil (1-\delta)(n_2 + 1)\rceil}{n_2 + 1} < (1 - \delta) + \frac{1}{n_2 + 1}.$$

**where**

(d) Given that the groupwise score distributions are continuous, the rank of $s_{\text{test}}$ among the scores in the corresponding group ($\mathcal{D}_1 \cup \{s_{\text{test}}\}$) is uniformly distributed. This makes the probability equal to the exact proportion of scores less than or equal to the quantile.

(e) By the property of the ceiling function $\lceil x \rceil < x + 1$

$\square$

### B.3. Proof of Proposition 4.2 ( Online Algorithm - Gaurantees )

First we present the following lemma that states that the thresholds $a_t$ and $b_t$ remain bounded at all time steps:

**Lemma B.1** (Parameter Boundedness). *Let $s(x,y) \in [0,1]$ be the non conformity scores. For any learning rate $\eta > 0$, the sequences $\{a_t\}$ and $\{b_t\}$ are bounded. Specifically, for all $t > 1$:*

$$b_t \in [-\eta\varepsilon, \ 1 + \eta(1 - \varepsilon)], \qquad a_t \in [-\eta\delta, \ 1 + \eta(1 - \delta)].$$

*Proof.* We prove the result for $b_t$; the proof for $a_t$ is symmetric.

Let $I_b = [-\eta\varepsilon, 1 + \eta(1 - \varepsilon)]$. We show by induction that once $b_t$ enters $I_b$, it never leaves.

First, since $b_1 \in [0, 1]$, the first update ensures $b_2 \in I_b$. Now, assume $b_t \in I_b$ for some $t > 1$.

**Upper Bound**: $b_{t+1}$ is maximized if the update is positive, which requires $\mathbf{1}\{s_t > b_t\} = 1$. This implies $s_t > b_t$, so $b_t$ must be less than $s_t \leq 1$. The update is $b_{t+1} = b_t + \eta(1 - \varepsilon)$. Since this increase only happens when $b_t < 1$, we have $b_{t+1} < 1 + \eta(1 - \varepsilon)$. If the update is negative, $b_{t+1} < b_t$, so it is also below the upper bound. Thus, $b_{t+1} \leq 1 + \eta(1 - \varepsilon)$.

**Lower Bound**: $b_{t+1}$ is minimized if the update is negative, which requires $\mathbf{1}\{s_t > b_t\} = 0$. This implies $s_t \leq b_t$, so $b_t$ must be greater than $s_t \geq 0$. The update is $b_{t+1} = b_t - \eta\varepsilon$. Since this decrease only happens when $b_t \geq 0$, we have $b_{t+1} \geq 0 - \eta\varepsilon = -\eta\varepsilon$. If the update is positive, $b_{t+1} > b_t$, so it is also above the lower bound. Thus, $b_{t+1} \geq -\eta\varepsilon$.

We have shown by induction that the parameters remain in their respective intervals for all $t > 1$. $\square$

Now first, lets restate the proposition

**Proposition B.2** (Finite-Sample Guarantees). *Let $N_1(T) = \sum_{t=1}^{T} \mathbf{1}\{Y_t \in H(X_t)\}$ and $N_2(T) = \sum_{t=1}^{T} \mathbf{1}\{Y_t \notin H(X_t)\}$. For any $T \geq 1$:*

$$\left| \frac{1}{N_1(T)} \sum_{t=1}^{T} \text{err}_t^{\text{in}} - \varepsilon \right| \leq \frac{1 + \eta \max(\varepsilon, 1 - \varepsilon)}{\eta N_1(T)}, \qquad \left| \frac{1}{N_2(T)} \sum_{t=1}^{T} \text{err}_t^{\text{out}} - \delta \right| \leq \frac{1 + \eta \max(\delta, 1 - \delta)}{\eta N_2(T)}.$$

We prove the first bound, and the second statement is symmetric. Let $I_1(T) = \{t \leq T \mid Y_t \in H(X_t)\}$ be the set of indices where the true label lies within the human proposed set. The number of such examples is $N_1(T) = |I_1(T)|$. As per algorithm, we only update the threshold $b_t$ for such points, and the update is given by:

$$b_{t+1} - b_t = \eta(\text{err}_t^{\text{in}} - \varepsilon)$$

where $err_{CH,t} = \mathbf{1}\{Y_t \notin C(X_t) \mid Y_t \in H(X_t)\}$ Note that this update only occurs to times $t \in I_{in}(T)$. If you sum over all relevant time steps where the update occurs we form a telescoping sum:

$$b_{T+1} - b_0 = \sum_{t \in I_1(T)} \eta(\text{err}_t^{\text{in}} - \varepsilon) = \eta \left( \sum_{t \in I_1(T)} \text{err}_t^{\text{in}} - \sum_{t \in I_1(T)} \varepsilon \right)$$

Since $err_t^1 = 0$ for all $t \notin I_1(T)$, we can expand the sum over all time steps and rearrange to get:

$$\frac{b_{T+1} - b_1}{\eta} = \sum_{t=1}^{T} \text{err}_t^{\text{in}} - \varepsilon N_1(T)$$

and rearranging again and taking the absolute value we obtain:

$$\left| \frac{1}{N_1(T)} \sum_{t=1}^{T} \mathrm{err}_t^{\mathrm{in}} - \varepsilon \right| = \left| \frac{b_{T+1} - b_1}{\eta N_1(T)} \right|$$

Using Lemma B.1, we can bound the numerator. The maximum value of $b_t$ is $1 + \eta(1 - \varepsilon)$ and the minimum is $-\eta\varepsilon$, which gives the bound $|b_{T+1} - b_1| \le 1 + \eta \max(1 - \varepsilon, \varepsilon)$. Substituting this directly we obtain our finite-sample inequality

$$\left| \frac{1}{N_1(T)} \sum_{t=1}^{T} \mathrm{err}_t^{\mathrm{in}} - \varepsilon \right| \le \frac{1 + \eta \max(1 - \varepsilon, \varepsilon)}{\eta N_1(T)}$$

# C. Additional Experimental Results

This section presents additional experimental results supporting the empirical findings in the main paper. We begin with a crowdsourcing study using real human prediction sets. We then provide supplementary results for each of the data modalities studied in the main text. While not included in the main text for readability, these results are fully consistent with the findings we discussed earlier. All additional experiments are complementary and serve to reinforce the core claims of the paper.

## C.1. Crowdsourcing Study

To complement the experiments in the main text, we conduct an additional study using prediction sets collected from real human participants. This experiment evaluates whether the empirical behavior of CUP persists when the human-proposed sets are obtained directly from crowdworkers rather than generated synthetically or derived from existing annotations.

**Recruitment and instructions.** The study was conducted on Prolific[1] with 50 participants. Participation was restricted to fluent English speakers with at least a 95% approval rate on prior Prolific studies. Participants were compensated at a rate of $12 USD/hour, and the median completion time was approximately 5 minutes for 20 annotation examples. Each participant was shown an image and asked to provide a set of plausible labels, corresponding to the human prediction set $H(x)$.

**Task design.** We constructed a pictorial reasoning task based on procedurally generated images. Each image contained a randomized collection of geometric shapes, including triangles, squares, and stars, placed at random positions on the canvas. The prediction task was to infer the relevant shape count from the image. The randomization of both the number and placement of shapes produced examples with varying levels of difficulty. Representative examples from the task are shown in Figure 4. Human prediction sets were collected directly from participant responses.

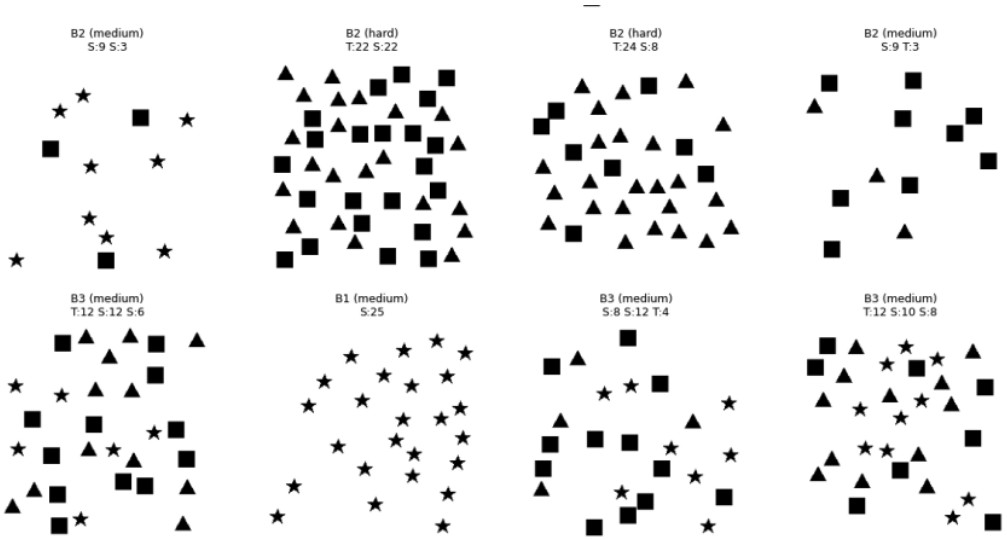

*Figure 4.* Representative examples from the crowdsourcing task. Images contain randomized configurations of triangles, squares, and stars, and participants were asked to provide a set of plausible labels based on the displayed image.

**AI model and evaluation.** For the AI component, we used Gemini 2.5 Flash-Lite (Gemini Team, Google, 2024) to produce a probability distribution over plausible labels for each example. These probabilities were used to compute the classification nonconformity scores used by CUP. Ground-truth labels were available from the procedural generation process and were used for calibration and evaluation. We evaluate the same three methods as in the main experiments: Human Alone, which returns the collected human set $H(x)$; AI Alone, which applies conformal prediction using only the AI scores; and CUP, which refines the human set using the proposed collaborative procedure.

---

[1]https://www.prolific.com/

**Offline results.** Table 4 reports offline results for several choices of $(\epsilon, 1 - \delta)$. Across all settings, CUP improves coverage over the human prediction sets while producing smaller sets than the AI-alone baseline at matched coverage. This mirrors the trends observed in the main experiments: CUP uses the AI to recover labels missed by the human while avoiding the larger sets required by the AI-only conformal baseline.

*Table 4.* Crowdsourcing task, offline setting. Entries report coverage / average set size (C/S).

| Human C/S | CUP C/S | $(\epsilon, 1 - \delta)$ | AI C/S |
|---|---|---|---|
| 0.494 / 3.000 | 0.645 / 2.715 | (0.01, 0.80) | 0.645 / 3.126 |
| 0.494 / 3.000 | 0.698 / 2.950 | (0.05, 0.60) | 0.698 / 3.217 |
| 0.494 / 3.000 | 0.809 / 3.664 | (0.05, 0.40) | 0.809 / 4.054 |
| 0.494 / 3.000 | 0.717 / 3.066 | (0.10, 0.50) | 0.717 / 3.295 |

**Online results.** We also evaluate CUP in the online setting, where thresholds are updated sequentially as examples arrive. Figure 5 shows cumulative coverage and cumulative mean set size for $(\epsilon, \delta) = (0.2, 0.5)$. CUP achieves higher cumulative coverage than the human baseline, reaching $64.35\%$ compared to $48.8\%$, while producing smaller sets than the AI-alone baseline at matched coverage: 2.72 for CUP compared to 3.22 for AI Alone. These results again support the same qualitative conclusion as the main experiments: CUP improves over the human prediction sets in coverage while using the human input to reduce set size relative to an AI-only conformal predictor.

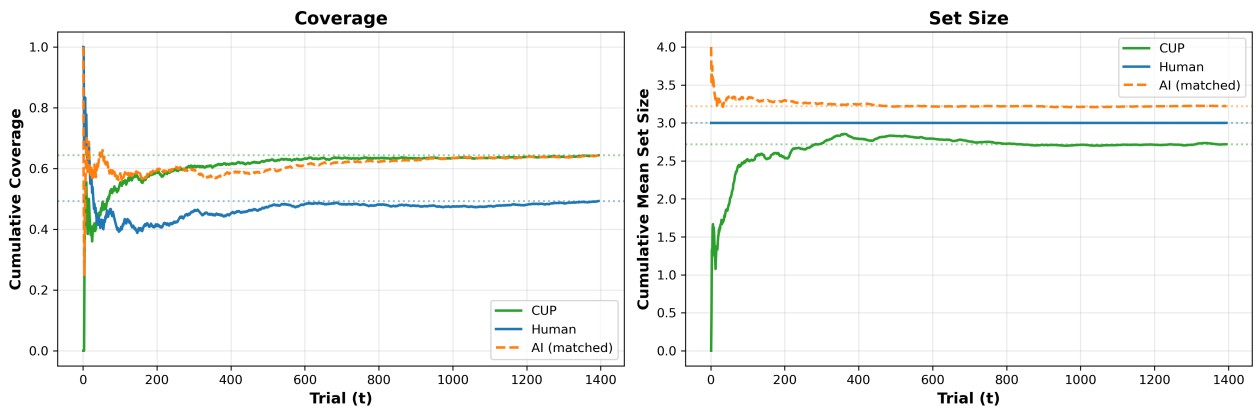

Figure R1: Online CUP on the crowdsourcing task. Left: CUP (64.35%) exceeds Human (48.8%) coverage.
Right: CUP produces smaller sets (2.72) than AI (3.22) at exact matched coverage.

*Figure 5.* Online CUP on the crowdsourcing task. Left: CUP achieves higher cumulative coverage than the human baseline. Right: CUP produces smaller cumulative mean set size than AI Alone at matched coverage.

### C.2. Classification: ImageNet-16H

Beyond the noise and human strategy shifts discussed in the main text, we also study a class label shift on ImageNet-16H. In this setting, calibration is performed on a restricted subset of classes, while evaluation takes place on a disjoint set of unseen classes. For instance, calibration may use only dog and cat images, while the test stream consists of bird images. This creates a particularly challenging shift, since the labels encountered at test time are entirely absent from calibration. We report the same metrics as in the main experiments. Figure 6 compares our adaptive online algorithm with the fixed baseline in terms of running coverage. As before, the adaptive method tracks the target levels closely, while the fixed baseline drifts and does not recover. Figure 7 then compares CUP against the Human-only and AI-only baselines. The pattern is consistent with other shifts: Relative to the human baseline, CUP raises coverage while also pruning incorrect labels from overly conservative sets, resulting in sharper and more informative predictions. Relative to AI alone, CUP achieves the same level of coverage with smaller sets, showing human input is being efficiently incorporated in the resulting prediction sets.

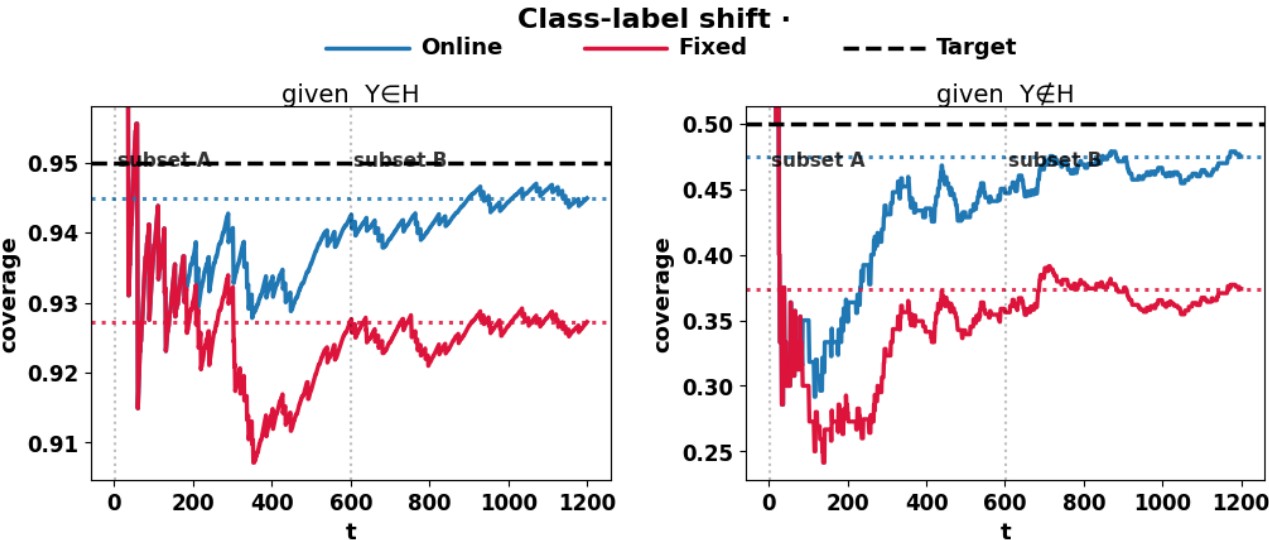

*Figure 6.* **Fixed vs. online CUP on ImageNet-16H under class label shift.** The online algorithm remains close to target coverage, while the fixed baseline drifts and fails to recover.

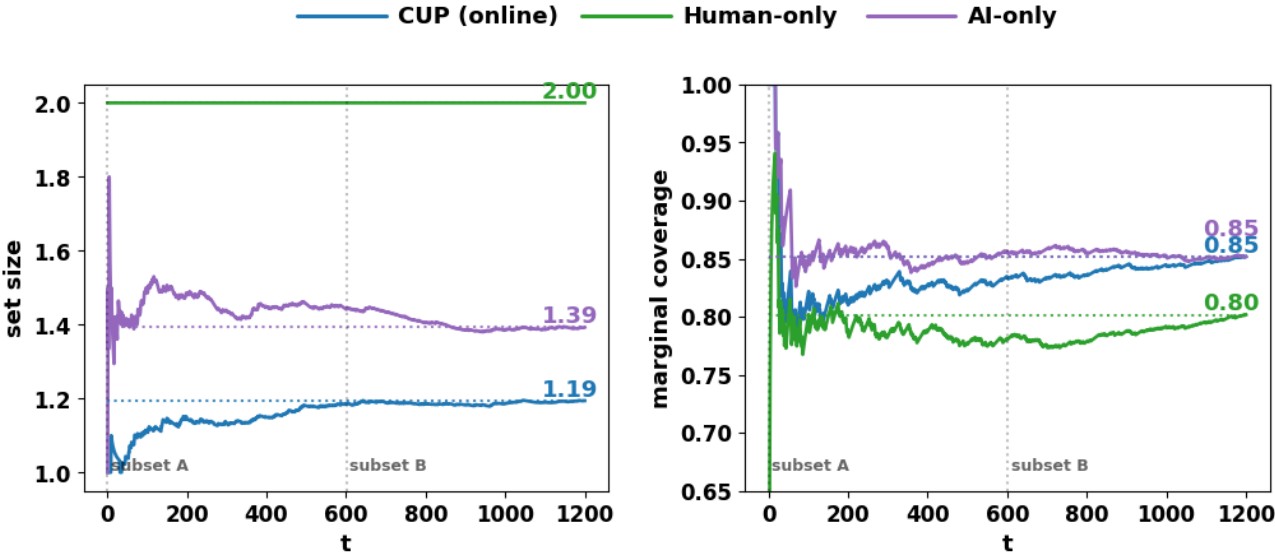

*Figure 7.* **Comparison against baselines on ImageNet-16H under class label shift.** CUP outperforms both Human-only and AI-only baselines, achieving higher coverage with smaller prediction sets.

### C.3. LLMs for medical diagnosis decision making

We begin with additional offline results on the DDXPlus dataset, exploring a wider range of calibration parameters $(\varepsilon, \delta)$. Table 5 reports coverage and set size for Human-only, AI-only, and CUP (ours) across both GPT-4o and GPT-5. These extra configurations make it possible to see how tuning the targets affects performance.

Across all reported settings, CUP improves upon the human baseline in at least one dimension, coverage or set size, with the magnitude of the gain depending on the specific $(\varepsilon, \delta)$ configuration. With the stronger model (GPT-5), CUP is often able to achieve simultaneous improvements in both dimensions: pruning away incorrect human labels while adding the correct one when needed, leading to higher coverage and smaller sets. With the weaker model (GPT-4o), coverage improvements are still observed, but they often come with larger set sizes, reflecting the model's more limited ability to prune. In all cases, however, CUP achieves smaller sets than AI alone at matched coverage levels, confirming that human input is effectively

incorporated.

| | Human | GPT-4o | | | GPT-5-mini | | |
|---|---|---|---|---|---|---|---|
| **Strategy** | **C/S** | **CUP C/S** | **$(\varepsilon, \delta)$** | **AI C/S** | **CUP C/S** | **$(\varepsilon, \delta)$** | **AI C/S** |
| Top-1 | 0.71 / 1.00 | 0.89 / 2.56 | (0.01, 0.65) | 0.88 / 4.58 | 0.87 / 1.27 | (0.02, 0.55) | 0.88 / 1.54 |
| Top-1 | 0.71 / 1.00 | 0.88 / 2.51 | (0.02, 0.65) | 0.88 / 4.40 | 0.88 / 1.36 | (0.01, 0.55) | 0.89 / 1.59 |
| Top-1 | 0.71 / 1.00 | 0.85 / 1.77 | (0.01, 0.50) | 0.85 / 3.69 | 0.85 / 1.19 | (0.02, 0.50) | 0.86 / 1.42 |
| Top-1 | 0.71 / 1.00 | 0.90 / 2.84 | (0.02, 0.70) | 0.88 / 4.64 | 0.91 / 1.59 | (0.02, 0.70) | 0.91 / 1.76 |
| Top-2 | 0.87 / 1.95 | 0.90 / 2.47 | (0.01, 0.30) | 0.88 / 4.57 | 0.95 / 2.31 | (0.01, 0.70) | 0.95 / 2.79 |
| Top-2 | 0.87 / 1.95 | 0.93 / 3.14 | (0.01, 0.45) | 0.90 / 9.12 | 0.94 / 1.73 | (0.02, 0.55) | 0.94 / 2.10 |
| Top-2 | 0.87 / 1.95 | 0.94 / 3.69 | (0.01, 0.55) | 0.93 / 22.41 | 0.91 / 1.51 | (0.02, 0.40) | 0.91 / 1.83 |
| Top-2 | 0.87 / 1.95 | 0.93 / 3.41 | (0.01, 0.50) | 0.91 / 13.55 | 0.93 / 1.65 | (0.02, 0.45) | 0.93 / 1.95 |

*Table 5.* **Additional configurations for offline setting: results on DDXPlus.** Human sets (shared across models) use a top-$k$ strategy. We compare the Human alone against CUP (ours) and AI Alone for two models side-by-side. Entries report *Coverage/Size* (C/S) with calibration parameters $(\varepsilon, \delta)$ shown for CUP.

We also report online results for DDXPlus under the same age-based distribution shift described in the main paper. Calibration is performed on younger patients, while testing proceeds on streams of older patients. These results, omitted from main text for readability, are included here for completeness.

Figure 8 shows the running coverage of CUP (online) compared to a fixed, non-adaptive variant. As in other modalities, the adaptive updates keep CUP close to the target levels throughout the stream, while the fixed baseline drifts and fails to recover.

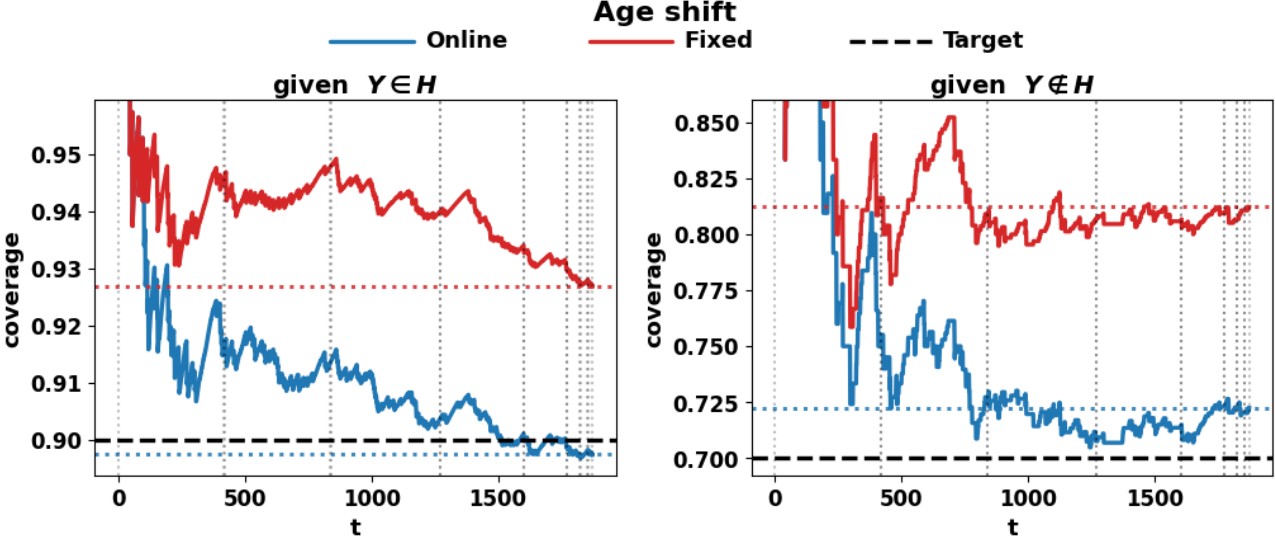

*Figure 8.* **Fixed vs. online CUP on DDXPlus under an age-based shift.** Same setting as in the main paper, included here for completeness. The online algorithm remains close to target coverage, while the fixed baseline drifts and fails to recover.

## C.4. Regression: Communities & Crime

As in the main paper for ImageNet, and in the appendix for LLMs, we evaluate CUP-online we against a fixed, non-adaptive variant in the regression setting with the UCI Communities & Crime dataset (Redmond, 2002) in Figure 9

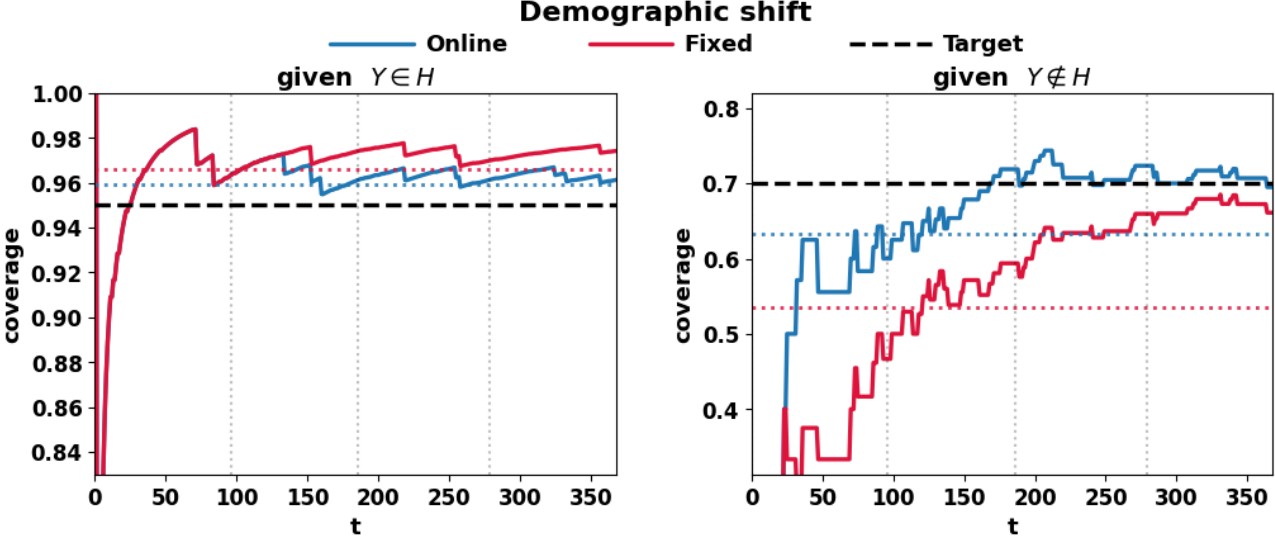

*Figure 9.* Fixed vs online CUP on Crime & Communitites dataset

