# OpenReview forum: "Human-AI Collaborative Uncertainty Quantification"
_ICML.cc/2026/Conference — ICML 2026 regular_

### Official Review · Reviewer_nvnS · 2026-03-06

**Soundness:** 3
**Presentation:** 3
**Significance:** 3
**Originality:** 3
**Overall Recommendation:** 5
**Confidence:** 3

**Summary:**

This paper proposes a collaborative framework for quantifying uncertainty through the lens of Conformal Prediction. The method aims to retrieve optimized conformal sets using human insights as starting points. First, the human proposes an initial set of plausible outcomes $H(x)$ and then the AI refines the proposal by outputting a prediction set $C(x,H(x))$ to complement the human input. This collaboration must satisfy low counterfactual harm and be complementary to the human annotation. Thus, the goal of the AI is to improve the trade-off between coverage and set size of the original conformal prediction.

Authors propose two methods: CUP-offline, where they assume access to a held-out calibration dataset that is exchangeable with the test data; and CUP-online, where data arrives sequentially, one sample at a time. The latter aims to fix the problems related to distribution shift of the offline method.

In order to apply this to both classification and regression, they use MCP and build upon Conformalized Quantile Regression (CQR), respectively.

Their results on Classification, LLM for medical diagnosis, and regression show that this collaborative systems improves both proposed baselines (Human-alone and AI-alone) by increasing coverage of the human and reducing set size of the AI-alone, improving the job done by the human annotator. Also, they show that the better the AI-model, the better the results achieved.

**Compliance With Llm Reviewing Policy:**

Affirmed.

**Final Justification:**

After considering the paper and the rebuttal, I view this work as a technically sound and novel contribution. My main concerns about the online setting, proxy human sets, and reproducibility were addressed satisfactorily during the discussion. Overall, the rebuttal increased my confidence in the paper and led me to a more positive final evaluation.

**Key Questions For Authors:**

- Q1: How does the human provide the prediction of the H(x)? Are they normal predictions like a label or top-k set or more like a confidence score?
- Q2: Apparently, the method always relies on labeled data, i.e. having access to the true label. This is normally not feasible in inference. How does the system know if $y_i\in H(x)$ or not?
- Q3: The change to ChatGPT-5-mini in Table 4 in the Appendix is intended or a mistake?
- Q4: How could this method be leveraged for OOD detection? We know that machines often struggle at detecting OOD, but humans could provide a better intuition for our uncertainty measures.
- Q5: In Figure 2 the limits of the x-axis for the marginal coverage are different than the rest of subplots. Is this intentional?
- Q6: During the adaptation of CUP-online, coverage may temporarily fail. The paper does not state how this transient behavior is. Do authors acknowledge that coverage might be violated while the online method is still adapting?
- Q7: In the results of e.g., Figure 2, the comparison of the human vs AI doesn’t seem optimally fair to me. While the human is giving top-k likely classes, the AI tries to find the smallest set that achieves a target coverage. These are not equivalent tasks, right? Conformal sets are coverage-calibrated but top-k predictions are not. In order to fix this, could conformal calibration be applied to the human sets?
- Q8: The medical diagnosis example seems to be shifted progressively. Does the method work under more abrupt shifts?
- Q9: Have you considered comparing against the human-conformal model system without  the alignment to isolate the specific improvement offered by the method (i.e. the impacts of the constraints)?

**Limitations:**

No. If the questions or pointed weaknesses are genuine (and not misunderstandings of the reviewer) authors could address some of those.

**Strengths And Weaknesses:**

Strengths:

- S1: The paper presents a technically sound and well-motivated formulation.
- S2: The notions of counterfactual harm and complementarity are natural and important for the problem formulation.
- S3: The online variant offers a significant improvement over the offline method without overcomplicating the method.
- S4: The paper is clearly written, well organized, and accessible even to readers with limited conformal prediction background.
- S5: The paper tackles an important problem at the intersection of human-AI collaboration and uncertainty quantification.
- S6: The work appears to be novel in the conformal prediction setting. Human-AI collaboration is not new, but I am not aware of similar approaches in conformal prediction. According to the authors, this is the first successful application of this kind, which strengthens the paper’s novelty.

Weaknesses:

- W1: In order for the method to work, the human’s behavior should be stable, there should not be a big shift in difficulty or population. The method relies on the assumption that the calibrated form of the validation part will stay the same for test (distribution shift), which is normally not guaranteed. CUP-online aims to overcome this problem, but this fix is not immediate: How many instances does the method need to converge again to a calibrated one?
- W2: The use of proxies instead of real Human annotations makes the experiments kind of “synthetic”. I am not sure if this is affecting the validity of the results.
- W3: The paper provides limited reproducibility evidence. Some implementation and experimental details seem insufficient to fully reproduce the reported results. Moreover, the results appear to be based on a single run rather than multiple random seeds, which may affect their robustness.
- W4: Presentation and format errors and errata:
    - In Section 4.1, line 214, there seems to be an issue with a parenthesis in the definition of the dataset.
    - At the end of Section 4.1, “relaxes allows” is awkward.
    - In line 248 (right column), the dot after “Human alone” should be a comma, right?
    - In line 281, there is a missing space in “large(e.g top-2)”, which should be “large (e.g top-2)”.
    - The title of Section 5.1 has a typo: “ImagetNet-16H” should be “ImageNet-16H”.
    - Figure 5 reference in C.1. should refer to Figure 4.
    - In Section C.3, line 1212, “we” appears to be duplicated.
    - In the same section, line 1233, there is a typo in the figure label: “Communitites” should be “Communities”.
    - $s(\cdot,\cdot)$ is called conformity instead of non-conformity a couple of times.

---

> ### Author Rebuttal · Authors · 2026-03-30
>
> Thank you for your careful and constructive review. We are glad you found the problem important and the theory sound. We address your concerns below.
>
> # On the convergence rate of the online algorithm (W1) and Q6
> Yes, temporary coverage violations may occur right after a shift. This is exactly what the Proposition 4.2 captures (you can apply it from any starting point). The result controls the running-average error, and the convergence rate appears explicitly in that bound. Importantly, the guarantee holds without assumptions on the stream, including the timing or size of shifts.
> **To better show the transient behavior, we now add finer-grained plots( Anonymous Link: https://imgur.com/a/qZPiNtn )**
>
> In particular, we report sliding-window average coverage, i.e., coverage computed over the most recent samples rather than the full history. This makes local failures and recovery after a shift directly visible: smaller windows react faster but are noisier, while larger windows are smoother but slower.
> We also include quantile-threshold traces over time. While the theorem controls error rates rather than the values of the thresholds themselves, these plots show that after a distribution shift the tracked quantiles enter a transient phase and then move toward new stable values.
>
> # Abrupt shift in the medical diagnosis task Q8
> **To address this, we now include an additional medical-diagnosis experiment with abrupt distribution shifts.( Anonymous Link: https://imgur.com/a/9MwGraG).** The same qualitative pattern holds: after the shift, the adaptive thresholds move quickly and recover toward the target conditional coverage substantially faster than a fixed baseline. We will include this result in the revision to make clear that the online procedure is not limited to gradual shifts.
>
> # On the Synthetic human sets W2
> We agree that real human sets strengthen real-world relevance. **To address this directly, we added a crowdsourcing experiment via Prolific with real human prediction sets, and the results are consistent with the main paper.** We will include these details in the revision. For the results summary please refer to our response to Reviewer KgC1.
>
> # On the reproducibility W3 and Presentation W4
> Thank you for pointing this out. All reported results are aggregated over multiple runs with different random seeds; they are not from a single run. We will clarify this explicitly, add more experimental details, and release the full code upon acceptance. We will also fix the formatting issues in the revision.
>
> # How Human sets are constructed Q1
> H(x) is a set of likely labels. In our experiments, we typically instantiate it as a top-k set; when probabilities are available behind the scenes, they are used only to form that set, and the algorithm itself only receives H(x), not human probabilities. In our crowdsourcing experiment, participants directly provide a list of likely labels.
>
> # Inference-time access to ground truth label Q2
> At test time, the algorithm does not use the ground-truth label y. It only uses the human set
> H(x): for labels in H(x), it applies the counterfactual-harm threshold; for labels outside H(x), it applies the complementarity threshold. The ground-truth label is used only during calibration to tune these two thresholds, not during inference.
>
> # OOD Q4
> We agree this is a promising direction. A natural signal is the behavior of the online quantile thresholds: when the stream changes, the thresholds often shift abruptly as the method adapts to maintain the target guarantees. In that sense, the threshold traces may serve as a useful signal of changes in the interaction. We agree this is a promising direction and will mention it more clearly in the discussion. The new threshold-evolution plots help illustrate why this connection is natural.
>
> # On the human-conformal model Q9, and Human vs AI sets Q7
> If calibrated human probabilities were available, one could design different algorithms. However, that is not the setting we study. From the algorithmic design viewpoint, our framework only receives a set from the human, while the AI side may provide richer uncertainty information. This is intentional as in practice, a human expert may provide a shortlist, ranking, or qualitative set of plausible options, but typically do not provide reliable calibrated probabilities.
>
> This also explains the comparison. The human top-k set is the smallest set the human believes captures the plausible labels, whereas the AI-only baseline is a conformal set calibrated to achieve a target coverage. These are therefore not meant to be identical procedures. Rather, CUP takes the human’s set-valued input as given and combines it with the AI’s uncertainty information in a principled way.
> In the experiments, human probabilities may exist behind the scenes to form proxies, but our method does not use those probabilities as input. We will clarify this distinction more explicitly in the introduction and experimental discussion.

---

> > ### Author Rebuttal · Reviewer_nvnS · 2026-04-03
> >
> > Thanks for the rebuttal. The clarification on CUP-online and the new plots are helpful. I will keep my score unchanged for now, both because Q3 and Q5 remain unanswered and because I would like to better understand the new Prolific experiment before revising my assessment. In particular, more detail on the protocol, participant instructions, annotation setup, and evaluation procedure would be valuable. The clarification that results are aggregated across multiple seeds is helpful as well. Beyond code release, I think the revision would benefit from a dedicated appendix section on reproducibility, including implementation details, practical issues encountered, and the analyses used to support the final hyperparameter choices.

---

> > > ### Author Response · Authors · 2026-04-04
> > >
> > > Thank you for your continued engagement and for the helpful follow-up questions. We are happy to provide details on the Prolific experiment, and also clarify Q3 and Q5, which we could not address in the original rebuttal due to space constraints.
> > >
> > > **Prolific Experiment Details**
> > >
> > > We are glad to provide full details on the crowdsourcing experiment and will include these in a dedicated appendix section in the revision.
> > >
> > > *Recruitment and Participant Instructions:* The study was conducted via Prolific. A total of 50 participants were recruited, with selection restricted to fluent English speakers with a minimum 95% approval rate on prior Prolific studies. Participants were compensated at $12 USD/hour; the median completion time was approximately 5 minutes for 20 annotation examples.
> > >
> > > *Task Design:* Participants were shown images and asked to provide a set of plausible labels (i.e., their prediction set H(x)). The task instructions presented at the beginning of the study are shown in https://imgur.com/a/6yd9gz0.
> > >
> > > *Dataset:* We constructed a dataset of synthetic images, each containing three types of shapes: triangles, squares, and stars. Images were procedurally generated with varying numbers of shapes at randomized positions across the canvas, yielding a range of task difficulties. Example images are shown in https://imgur.com/a/vUrQADX.
> > >
> > > *Annotation and Evaluation Procedure:* Human prediction sets H(x) were collected directly from participants. The AI (Gemini 2.5 Flash) was asked to report a probability distribution over plausible shape counts; these distributions were used as input to the CUP algorithm. Ground-truth shape counts were available for calibration and evaluation.
> > >
> > > **Reproducibility Appendix**
> > > We agree that a dedicated appendix section on reproducibility would strengthen the paper. The revision will include full implementation details, practical issues encountered during development, and the analyses supporting our final hyperparameter choices, in addition to the planned code release upon acceptance.
> > >
> > > **Q3: GPT-5-mini in Table 4**
> > > This was a typo. The results correspond to GPT-5, consistent with the model described in the paragraph that explains Table 4 in the appendix. We thank you for catching this and will correct it in the revision.
> > >
> > > **Q5: X-axis limits in Figure 2**
> > > Thank you for the careful observation. Upon revisiting the figure, we noticed a rendering inconsistency in how the marginal coverage panel was displayed. We have replotted the figure so that both panels share a consistent x-axis, making the comparison cleaner and easier to read. The qualitative conclusions are unchanged; CUP maintains its advantages in both coverage and set size over both baselines throughout the full stream. Updated plots are available here: https://imgur.com/a/XcUwmIo

---

### Official Review · Reviewer_X8Ca · 2026-03-11

**Soundness:** 3
**Presentation:** 3
**Significance:** 3
**Originality:** 3
**Overall Recommendation:** 5
**Confidence:** 3

**Summary:**

The paper introduces a method to quantify uncertainty by combining human judgements and AI predictions. The proposed approach provides prediction sets that are constrained to two principles, i.e., counterfactual harm and complementarity, which ensure that the obtained prediction sets improve over single prediction sets. The authors further provide the optimal collaborative prediction set and show that it can be obtained by thresholding a single score. Moreover, they offer distribution free guarantees for both an offline and online algorithm implementing the optimal strategy. Empirical results show the effectiveness of the method.

**Compliance With Llm Reviewing Policy:**

Affirmed.

**Final Justification:**

I think this is an interesting work, worth acceptance.

**Key Questions For Authors:**

I have a couple of questions:

1. I might have missed it, but it is not entirely clear to me how $(\varepsilon, \delta)$ are picked in practice. Can you clarify it?
2. I noticed that in Table 1, it seems that the set sizes are larger for lower noise images than for larger noise images. I guess this is due to different $(\varepsilon, \delta)$, but can you provide the same results using the same values for both experiments?

**Limitations:**

Limitations are not discussed. I would encourage the authors to provide a list of limitations in the final version.

**Strengths And Weaknesses:**

The main strengths of the work are:

1. **Theoretically sound approach.** The work is theoretically sound, and the proofs seem mostly correct to me.
2. **Interesting new perspective to combine human and AI.** I think the proposed approach offers a nice new perspective on human-AI decision-making.
3. **Relevant and potentially impactful approach.** The work has the potential to be impactful in the human-AI decision-making community.
4. **Generally well-written and motivated.** I enjoyed reading the work, and I think the authors present their contributions clearly.

Overall, I think this is a solid work. Thus, I would like to point out a few (minor) shortcomings and provide some food for thought:

1. **Be clear in Theorem 2.1 statement.** I noticed that in the proof, you rely on an LP relaxation of the problem. I would be explicit about it even in the main statement.
2. **Extending related work with other references on uncertainty in AI/ML.** I think the authors cover the conformal prediction literature quite extensively. I would suggest the authors to have a look at [1,2,3] to extend the related work covering also some other aspects of the uncertainty quantification literature, e.g., epistemic vs aleatoric uncertainty, calibration and abstaining ML models;
3. **Presentation of Figures can be improved.** I guess this is mostly due to the lack of space, but figures are very small at the moment, making it hard to read them. I would suggest increasing their size in case of acceptance.
4. **Anytime-valid guarantees.** A few recent works in conformal prediction suggest the usage of e-values to improve the guarantees of conformal predictors (see e.g., [4,5]). I guess exploiting similar ideas could provide stronger guarantees (even for future works).



[1] - Hüllermeier, E., & Waegeman, W. (2021). Aleatoric and epistemic uncertainty in machine learning: An introduction to concepts and methods. Machine learning, 110(3), 457-506.

[2] - Silva Filho, T., Song, H., Perello-Nieto, M., Santos-Rodriguez, R., Kull, M., & Flach, P. (2023). Classifier calibration: a survey on how to assess and improve predicted class probabilities. Machine Learning, 112(9), 3211-3260.

[3] - Ruggieri, S., & Pugnana, A. (2025). Things machine learning models know that they don’t know. AAAI

[4] - Gauthier, E., Bach, F., & Jordan, M. I. (2026). Adaptive coverage policies in conformal prediction. To appear in AISTATS 2026.

[5] - Gauthier, E., Bach, F., & Jordan, M. I. (2025) Backward Conformal Prediction. NeurIPS.

---

> ### Author Rebuttal · Authors · 2026-03-30
>
> We thank the reviewer for the clear and encouraging feedback. We are glad that you found the paper theoretically sound, the problem setting important and well motivated and the contributions clearly articulated. We address each point below:
>
> # On Theorem 2.1
> Thank you for pointing this out. We agree the theorem statement should be clearer. **The LP formulation used in the proof is not a loose relaxation.** It is a **lossless reformulation of the original problem** and the **theorem recovers the exact solution** to the original optimization. We will revise the proof discussion to make this more bold in the beginning of the proof.
>
> # On related work beyond CP
> Thank you for your suggestion. We agree that the related work can better situate the paper within the broader uncertainty quantification literature. In the revision we will include all five suggested references [1,2,3,4,5], and explain the discussion to explicitly capture related perspective on epistemic vs aleatoric uncertainty, calibration, and abstention.
>
> # Anytime-valid Gaurantees
> We agree that this is indeed a promising direction. E-value and anytime-valid ideas can complement our framework and lead to algorithms with potentially stronger online guarantees, and we will mention this in the future work.
>
> # Choosing 𝜖,𝛿
> These are user-specified risk tolerances, analogous to choosing a target miscoverage level in conformal prediction. In practice, they can be set based on domain risk preferences, swept to study the tradeoff with set size, or chosen based on downstream decision requirements. In our experiments, we report regimes of ϵ and 𝛿 where CUP outperforms the human baseline in at least one metric, and often both coverage and set size. Additional configurations are provided in the appendix, which show how more permissive or more restrictive choices of 𝜖, 𝛿 affect the resulting coverage and set-size tradeoff.
>
> # On Table 1
> The difference is not only due to different target coverage levels for CUP, but also because the human baseline itself operates at different coverage regimes across noise levels under the same top-k strategy.In particular, for lower-noise images, the human top-1 set already achieves substantially higher marginal coverage than in the higher-noise setting. Our reported ϵ,δ values are chosen to show regimes where CUP improves over the human’s achieved coverage and the resulting set size. Because of this, forcing a matched target across noise levels would not be very meaningful: for example, at low noise the human top-1 baseline is already around 92% coverage, so reporting a 90% target there would not reflect the relevant tradeoff. To make this clearer, we will add more fine-grained 𝜖,𝛿 configurations in the appendix to show the coverage–set size tradeoff within each noise level.
>
> # On Limitations
> We agree that the paper should discuss its limitations explicitly, and we will add a limitations paragraph in the final version. This omission was largely due to space constraints. In particular, we will discuss the limitation that the collaboration is restricted to a single round of communication, as opposed to multi-round back-and-forth refinement, as well as the marginal nature of our guarantees for both counterfactual harm and complementarity. These limitations also point to interesting directions for future work.

---

> > ### Author Rebuttal · Reviewer_X8Ca · 2026-04-02
> >
> > I thank the authors for their answers. Hence, I keep my already positive score.

---

### Official Review · Reviewer_uH8A · 2026-03-12

**Soundness:** 2
**Presentation:** 3
**Significance:** 2
**Originality:** 2
**Overall Recommendation:** 3
**Confidence:** 1

**Summary:**

The paper proposes a framework for human-AI collaboration under uncertainty. In this framework, an AI system adjusts a human expert’s prediction set while ensuring it does not worsen correct human decisions and helps recover correct outcomes that the human missed. The authors show that the optimal solution follows a two-threshold rule and develop distribution-free calibration algorithms that adapt to changing data or human behavior.


Due to personal circumstances beyond my control, I was unfortunately unable to complete a thorough and careful review of the submission that it deserves. While I have requested to be removed from reviwers list, I have not been formally removed from this assignment. I am providing comments and offering limited feedback. However, I would like to emphasize that these comments, if any, should be considered preliminary observations rather than a comprehensive evaluation. Therefore, I emphasize, they should not be taken into account when determining the paper’s score or overall assessment by the AC, and should not be interpreted as my final evaluation.

**Compliance With Llm Reviewing Policy:**

Affirmed.

**Ethical Review Concerns:**

It seems human-subject experiments are used for evaluatution but I could not find any information on IRB approval/compliance. Any human-subjects experiment that yields generalizable results should go through the IRB for ethical and compliance review. And discussion of the experimental protocol, participant recruitment/target population, etc.

**Ethics Expertise Needed:**

["Responsible Research Practice (e.g., IRB, documentation, research ethics)"]

**Key Questions For Authors:**

Can the authors discuss which theorem was unknown before, and not just a reformulation of an existing theorem for a different context?

Here it is uses concepts in conformal prediction/set; can the authors discuss the connection between their work and different sources of uncertainties?

**Limitations:**

See above.

**Strengths And Weaknesses:**

The paper seems original and interesting, though the lack of comprehensive experimentation puts it in the realm of almost purely theoretical work. For a theoretical work, it mostly uses and repurposes existing conformal prediction results in a different context; therefore, the novelty of this work for the ICML bar is questionable.

Specifically, the empirical evaluation and benchmarking are very limited; for instance, how the proposed framework/method compares with the methods that consider uncertainty quantification for developing a collaboration strategy (e.g., https://www.nature.com/articles/s41746-023-00797-9). While I am not an expert in this topic, there has been a lot of work on Human-AI collaboration, and the lack of benchmarking for highly experimental work is unsatisfying.

Another observation is that there is no distinction between epistemic vs altruistic uncertainties; is such a distinction not useful in this framework? Why not? If this is useful, how is the distinction handled?

---

> ### Author Rebuttal · Authors · 2026-03-30
>
> We thank the reviewer for taking the time to provide feedback despite the difficult circumstances. We are glad the reviewer found the paper original and interesting. We address the main concerns below.
>
> # On Empirical Evaluations
> In addition to the three modalities already included in the submission, we have now added a crowdsourcing study with real human participants via Prolific. For a summary of the results please refer to the response to reviewer KgC1. In this study, participants directly provide prediction sets on a pictorial reasoning task. The results are consistent with the main paper: CUP improves over the human-alone baseline in coverage while maintaining smaller set sizes than the AI-alone baseline. We will include the full task design, participant details, and results in the revised version.
>
> More broadly, the current evaluation already spans three distinct modalities: image classification, medical diagnosis, and regression, which we believe is important evidence that the framework is not tied to a single domain. The added crowdsourcing study further strengthens the real-world applicability of the method.
>
> # On benchmarking against prior human-AI collaboration methods
> To the best of our knowledge, there is no prior method that addresses the same problem we study: constructing a collaborative prediction set that explicitly controls counterfactual harm and complementarity relative to a human-provided set, without modeling the human.
>
> The closest prior works are the human-AI prediction-set methods discussed in our related work. However, those methods study a different problem: the AI constructs a prediction set and presents it to the human, and success is measured by the human’s final decision quality after seeing that set. As a result, they either explicitly or implicitly rely on a model of how the human uses set-valued advice. Our framework is different in three key ways:
>
> 1. the human provides a set first and the AI refines it,
> 2. the output is a collaborative set, not a downstream human action, and
> 3. the guarantees are stated directly on that set, through counterfactual harm and complementarity, without any assumption on human behavior.
>
> For example, in the closest work of Straitouri et al. (2023), the machine side is essentially an AI-generated conformal prediction set shown to the human. In that sense, our AI-only baseline already captures the relevant prediction-set construction part of that paradigm. What their framework adds is a model of how the human responds to that set and an objective based on the resulting human accuracy. That is not the object we optimize.
>
> This is also why a comparison to the linked medical HAI paper is not straightforward. That work studies a different collaboration mechanism, does not construct collaborative sets, and does not enforce counterfactual harm or complementarity. We agree, however, that the paper should explain these distinctions more clearly, and we will make sure to include this paper in the related work and clarify these points explicitly.
>
>
> # On originality and role of CP
> We appreciate this concern and agree that some algorithmic components build on standard conformal prediction tools. However, the main novelty lies in conceptual framing of human-AI collaboration. The core contributions are:
> 1. A new formulation of human-AI collaboration through set-valued prediction, centered on the two collaboration principles of counterfactual harm and complementarity.
> 2. A new optimization problem with conditional constraints defined relative to the human set.
> 3. A new structural result: the optimal collaborative prediction set has a two-threshold form.
> 4. Finite-sample offline and online procedures that calibrate these constraints without modeling the human behavior.
>
> In particular, Theorem 2.1 is not a restatement of a standard conformal result. The optimization problem itself is new, because the constraints are conditional on the human set and encode the two collaboration principles. The resulting two-threshold structure is intuitive in hindsight, but it is a consequence of a new analysis in this setting. We will make this distinction more explicit in the revision.
>
> # Other notions of uncertainty and epistemic vs aleatoric uncertainty
> Our framework does not require separating epistemic from aleatoric uncertainty. It only assumes a score function, and that score may reflect epistemic, aleatoric, or mixed uncertainty. We use CP because our object is a prediction set with finite-sample coverage control, which matches our CH/COMP constraints. Thus, our framework is complementary to broader UQ methods. Different UQ signals and estimates can be used to define the score, while CP is the calibration layer that gives set-valued guarantees. We will clarify these distinctions in the revision.

---

> > ### Author Rebuttal · Reviewer_uH8A · 2026-04-02
> >
> > My concern about the lack of proper evaluation is not properly addressed here. I can see that other works use different principles for collaboration strategies; however, one can implement those strategies and, given the objectives of this work, assess how well they are doing. If they achieve the same or superior performance (given the criteria considered in this work), even though their setup, assumptions, or objectives are very different, I am not sure the proposed strategy here would be justified.

---

### Official Review · Reviewer_KgC1 · 2026-03-13

**Soundness:** 3
**Presentation:** 3
**Significance:** 3
**Originality:** 3
**Overall Recommendation:** 5
**Confidence:** 3

**Summary:**

This paper proposes a framework for human-AI collaborative uncertainty quantification in which a human expert first proposes a set of plausible labels, and an AI system refines that set to produce a final prediction set. The framework is motivated by two design principles: counterfactual harm, requiring that the AI not degrade correct human predictions, and complementarity, requiring that the AI recover correct outcomes that the human missed. These principles are formalized as conditional coverage constraints in an optimization problem that minimizes expected set size while maintaining guarantees about performance relative to the human proposal. The authors show that the optimal collaborative prediction set has a two-threshold structure over a single score function and propose practical finite-sample algorithms in both offline and online settings. Empirical results on image classification (ImageNet-16H), medical diagnosis using LLMs (DDXPlus), and regression (Communities & Crime) demonstrate improved coverage–size tradeoffs compared to human-only and AI-only baselines.

**Compliance With Llm Reviewing Policy:**

Affirmed.

**Final Justification:**

Following updates from the authors int eh rebuttal period, including responses to other reviewers, I have updated my score to "5: Accept". They addressed my concerns to the extent possible within the rebuttal period, and have included additional details and experiments that, when added to the camera ready version, increase originality value of the paper (I now rate this at 3).

**Key Questions For Authors:**

NA

**Limitations:**

yes

**Strengths And Weaknesses:**

Soundness: The paper presents a mathematically coherent framework. The HACO formulation and the two-threshold optimality result are well motivated, and finite-sample guarantees are provided. The empirical evaluation tests the method across three modalities (image classification, medical diagnosis with LLMs, tabular data regression), providing good evidence of the generality of the approach. However, a limitation is that in two of the three experimental settings the human sets are synthetically constructed. While these approximations are reasonable proxies, they do not fully capture the complexities of real human decision-making. As a result, the experiments demonstrate improvements relative to simulated human sets, not necessarily real expert interaction. The experiments compare only against human-only or AI-only, but not against other prediction-set collaboration approaches discussed in related work. Since the main contribution concerns human-AI collaboration with prediction sets, including such comparisons would strengthen the empirical validation.

Presentation: The paper is well structured. The schematic illustration in Figure 1 effectively conveys the relationship between counterfactual harm and complementarity. The experimental section is also organized clearly by modality (classification, medical text, regression), which helps demonstrate the generality of the approach. However, the distinction between this framework and existing human-AI prediction-set systems could be clearer. Section 1.1 discusses related work but primarily contrasts with “learning to defer” methods rather than with other prediction-set collaboration approaches. The distinction between these approaches (e.g. in Straitouri et al. (2023), Babbar et al. (2022), De Toni et al. (2024)) and the present framework is only partially articulated. The paper states that prior work studies “how humans use AI-provided sets” whereas this work constructs a joint prediction set algorithmically (Section 1.1). Including empirical comparisons against at least one of these prediction-set collaboration methods would clarify how the proposed framework differs in practice.

Significance: The problem of combining human expertise and machine learning predictions under uncertainty is important and relevant for any area where humans may use AI decision support, especially high-stakes scenarios with uncertainty. The explicit formalization of “counterfactual harm” is also valuable from a trust and safety perspective.

The experiments demonstrate that collaborative prediction sets can outperform both human-only and AI-only baselines. For example, in the ImageNet-16H classification experiment, the author's CUP algorithm achieves ~10 percentage-point coverage improvement while also reducing set size by ~25 %, which is a meaningful improvement. However, while improvements are consistent (as predicted by the theory), the magnitude of improvements is variable and often smaller. Additionally, several of the experiments rely on simulated human responses rather than real human data, so the significance of the claims to the actual real-world settings is limited.

Originality: The paper introduces a clear conceptual framework for collaborative prediction sets centered on two principles (counterfactual harm and complementarity). The theoretical result showing that the optimal collaborative set takes a two-threshold form is also appealing. The algorithmic components largely build on existing conformal prediction methods: the classification score uses the standard non-conformity score, the regression approach extends Conformalized Quantile Regression, and the offline calibration uses standard conformal quantile procedures. The method could be seen as an extension of conformal calibration to a two-constraint setting rather than a fundamentally new paradigm.

---

> ### Author Rebuttal · Authors · 2026-03-30
>
> We thank the reviewer for their detailed and constructive feedback. We are glad they found the framework well-structured, the problem high-impact, and the theoretical results appealing. We address each of your concerns below.
>
> # On the use of Synthetic human sets
> To directly address this concern, we conducted a **crowdsourcing study via Prolific with real human participants**. Participants were instructed to produce prediction sets on a pictorial reasoning task. The results confirm our theoretical guarantees and are consistent with the findings reported across the three original datasets. Across all settings, CUP consistently achieves higher coverage than the human alone at smaller set sizes, while also reducing set size relative to the AI alone. A summary of Offline-setting results are shown below and online results are available via the anonymous link (https://imgur.com/a/w86mwwP). Full task design, participant details, and additional results will appear in the camera-ready version.
> ***offline results:***
> | Human C/S | CUP C/S | (ε, 1−δ) | AI C/S |
> |---|---|---|---|
> | 0.494 / 3.000 | 0.645 / 2.715 | (0.01, 0.80) | 0.645 / 3.126 |
> | 0.494 / 3.000 | 0.698 / 2.950 | (0.05, 0.60) | 0.698 / 3.217 |
> | 0.494 / 3.000 | 0.809 / 3.664 | (0.05, 0.40) | 0.809 / 4.054 |
> | 0.494 / 3.000 | 0.717 / 3.066 | (0.10, 0.50) | 0.717 / 3.295 |
>
> We also note that datasets with real expert prediction sets remain scarce across modalities, and synthetic sets were simply a means to extend our evaluation to settings where such data is not yet available to the community. To that end, we will make our crowdsourcing task and collected data publicly available in case of acceptance.
>
> # On the distinction from prior HAI prediction-set approaches.
> We agree that the distinction from prior HAI prediction-set work should be made more explicit. The works mentioned, study a genuinely interesting but different problem. In their methods, the AI constructs a prediction set and presents it to the human, and the quantity of interest is the human’s final decision quality after receiving that set. As a result, these approaches either explicitly or implicitly rely on a model of how the human interprets and acts on set-valued advice.
>
> Our framework differs in several concrete ways. First, the output of our method is a final collaborative prediction set, not a downstream human choice. Second, our guarantees are stated directly on that set, through the two collaboration criteria of counterfactual harm and complementarity, and they hold without any assumption on human behavior. In contrast, prior prediction-set collaboration methods do not explicitly control these quantities, since they are not designed to construct sets that satisfy such constraints. For example, in the closest prior work of Straitouri et al. (2023), the AI side is a conformal prediction set that is then shown to the human, after which performance is evaluated through the human’s final answer. In that sense, our AI-only baseline already captures the relevant prediction-set construction component of that paradigm. Their framework adds a model of how the human responds to that set and an optimization objective based on the resulting human accuracy. Our setting does not model that downstream human response, because that is not the object we seek to optimize; instead, we seek to construct a collaborative set that directly satisfies counterfactual harm and complementarity constraints without any assumption on the human behavior.
>
> For these reasons, a direct empirical comparison is not straightforward. The two approaches are designed for different goals and operate through different pipelines, so they are not naturally applied within a common experimental setup.
>
> # On Originality and the relationship to existing conformal prediction methods
> We appreciate the reviewer's careful characterization here. We agree that the individual algorithmic components ( e.g non-conformity scores, CQR ) build on well-established tools. The two-constraint calibration structure is the key technical novelty at the algorithmic level, and we do not claim otherwise.
> That said, we believe the primary contribution is conceptual rather than purely algorithmic. The notions of counterfactual harm and complementarity had not previously been cast as formal statistical coverage constraints, and the resulting two-threshold optimality is a non-trivial consequence of this formulation. More broadly, we view the main contribution as providing a principled and assumption-free statistical foundation for human-AI collaborative prediction, one that is agnostic to both the human's behavior and the AI's internal mechanism. We will clarify this framing more explicitly in the introduction and contribution statement of the revised manuscript.

---

> > ### Author Rebuttal · Reviewer_KgC1 · 2026-04-06
> >
> > Thanks for the detailed reply and for the additional experiment report - I have also looked a the additional details on this experiment provided to another reviewer's comments. While the experiment setup is very basic, this is reasonable in the time available in the rebuttal period - but therefore only partly addresses my comment that the experiments do not capture "real expert interaction" with an AI system.
> >
> > I think the authors' explanation of their task setup as distinct from prior approaches, and therefore not directly comparable in a meaningful sense, is technically correct, and I appreciate the explanation here and in response to other reviewers. However, this seems to limit the practical significance of this setup: in most cases, ultimately in most decision support tasks, a single decision will need to be made based on the human + model output (e.g. as in the setup in Straitouri et al. 2023). But the author's framework ends with a prediction set, rather than a single decision. Could the authors comment on this apparent limitation, and how they view their framework being used practically in real-world settings? Additional detail on the practical implementation of this framework could increase the potential significance of this work beyond the nice theoretical results.

---

> > > ### Author Response · Authors · 2026-04-07
> > >
> > > Thank you, we are glad that most of your concerns have been addressed through our rebuttal, and appreciate you raising this remaining point, as it gives us the opportunity to clarify the practical scope of our framework.
> > >
> > > **On practicality of the framework:** You are right that in most decision support tasks, a single decision ultimately needs to be made. However, we view uncertainty quantification and decision making as two distinct but complementary components of an end-to-end system. Uncertainty quantification asks: given the available information, what is a principled representation of what the true label or state could be? Decision making then asks: given this representation, what action should be taken? Studying UQ in isolation is itself a meaningful and well-established research agenda and this is precisely how the conformal prediction community has evolved, developing rigorous tools for UQ with prediction sets independently of any specific downstream decision rule.
> > >
> > > That said, there is a growing body of work connecting conformal prediction sets to optimal downstream decision making [1,2]. Crucially, our collaborative prediction sets can serve as direct inputs to any such decision-making method, including those that map sets to actions in a principled way. In this sense, our work builds a foundation for human-AI collaborative UQ that can be plugged into any downstream decision pipeline.
> > >
> > > This also clarifies an important distinction from prior work. In prior frameworks like (Straitouri et al.), the human does not explicitly participate in the UQ step. The role of the human is limited to selecting a final action from the AI-provided set, rather than actively contributing to the construction of the uncertainty representation itself. In our framework, the collaboration happens at the UQ level itself, which we believe is perhaps precisely where it is most valuable. The AI revises and builds upon the human's proposed set to form a collaborative uncertainty representation that reflects both human expertise and AI capabilities. The resulting collaborative set can then be fed to any decision-making method, making our framework a flexible foundation for the UQ component.
> > >
> > > **On the concern that the crowdsourcing experiment does not fully capture real expert interaction:** We appreciate the reviewer's understanding on this point. Collecting real expert prediction sets at scale is prohibitively expensive and logistically challenging, and to our knowledge no benchmark currently exists that provides such data across multiple modalities. We are glad that our ImageNet-16H experiment (included in the original submission) with real human annotations and our crowdsourcing study were found meaningful, and we will expand the experimental details and discussion in the camera-ready version. We also plan to publicly release our crowdsourcing dataset upon acceptance, which we hope can serve as a step toward building richer benchmarks for human-AI collaboration research.
> > >
> > >
> > > [1] Kiyani, Pappas, Roth & Hassani, "Decision Theoretic Foundations for Conformal Prediction: Optimal Uncertainty Quantification for Risk-Averse Agents," ICML, 2025.
> > >
> > > [2] Wang & Dobriban, "Optimal Decision-Making Based on Prediction Sets," arXiv:2602.00989, 2026.

---

### Decision · Program_Chairs · 2026-04-30

**Decision:**

Accept (regular)

**Comment:**

This work studies uncertainty quantification in the setting where human and AI collaborate to produce prediction sets. Specifically, the AI is trained to modify the human's prediction sets under two separate coverage-rate constraints conditioned on whether or not the human's prediction set covers the true label. The reviewers appreciate the soundness, originality, clarity, and significance of the submission. I recommend acceptance.